# A mathematical model describing the localization and spread of influenza A virus infection within the human respiratory tract

Christian Quirouette[1], Nada P. Younis[1], Micaela B. Reddy[2], Catherine A. A. Beauchemin[1,3]*

**1** Department of Physics, Ryerson University, Toronto, Ontario, Canada, **2** Array BioPharma Inc., Boulder, Colorado, United States of America, **3** Interdisciplinary Theoretical and Mathematical Sciences (iTHEMS), RIKEN, Wako, Japan

* cbeau@ryerson.ca

**Data Availability Statement:** All relevant data are within the manuscript and its Supporting Information files.

## Abstract

Within the human respiratory tract (HRT), virus diffuses through the periciliary fluid (PCF) bathing the epithelium. But virus also undergoes advection: as the mucus layer sitting atop the PCF is pushed along by the ciliated cell's beating cilia, the PCF and its virus content are also pushed along, upwards towards the nose and mouth. While many mathematical models (MMs) have described the course of influenza A virus (IAV) infections in vivo, none have considered the impact of both diffusion and advection on the kinetics and localization of the infection. The MM herein represents the HRT as a one-dimensional track extending from the nose down towards the lower HRT, wherein stationary cells interact with IAV which moves within (diffusion) and along with (advection) the PCF. Diffusion was found to be negligible in the presence of advection which effectively sweeps away IAV, preventing infection from disseminating below the depth at which virus first deposits. Higher virus production rates (10-fold) are required at higher advection speeds (40 μm/s) to maintain equivalent infection severity and timing. Because virus is entrained upwards, upper parts of the HRT see more virus than lower parts. As such, infection peaks and resolves faster in the upper than in the lower HRT, making it appear as though infection progresses from the upper towards the lower HRT, as reported in mice. When the spatial MM is expanded to include cellular regeneration and an immune response, it reproduces tissue damage levels reported in patients. It also captures the kinetics of seasonal and avian IAV infections, via parameter changes consistent with reported differences between these strains, enabling comparison of their treatment with antivirals. This new MM offers a convenient and unique platform from which to study the localization and spread of respiratory viral infections within the HRT.

## Author summary

This work proposes a new way to think about and model the dissemination of an influenza A virus (IAV) infection within the human respiratory tract (HRT). The computational model takes into account the physiological environment in which the infection

**Funding:** This work was supported in part by Discovery Grant 355837-2013 (CAAB) from the Natural Sciences and Engineering Research Council of Canada (www.nserc-crsng.gc.ca), Early Researcher Award ER13-09-040 (CAAB) from the Ministry of Research and Innovation of the Government of Ontario (www.ontario.ca/page/early-researcher-awards), and by Interdisciplinary Theoretical and Mathematical Sciences (iTHEMS, ithems.riken.jp) at RIKEN (CAAB). The funders had no role in study design, data collection and analysis, or decision to publish.

**Competing interests:** I have read the journal's policy and the authors of this manuscript have the following competing interests: CAAB received financial support in the form of a research contract and a consultancy fee from F. Hoffmann-La Roche Ltd. in the early stages of this project. MBR was employed by F. Hoffmann-La Roche Ltd. when first participating in this work, and is currently employed by Array BioPharma Inc.

takes place by representing the HRT spatially in one dimension (depth), and by incorporating the effect of virion diffusion within the periciliary fluid that bathes infectable cells, and the remarkable physiological barrier that is the mucus escalator, sweeping virus upwards. Cell regeneration, the immune response, and infection with human vs avian IAV strains are explored in this spatial context. The numerical efficiency of this model, compared to agent-based models, makes it an attractive alternative to model respiratory virus infections in vivo.

## Introduction

Mathematical models (MMs) of influenza A virus (IAV) infection kinetics are mainly based on ordinary differential equations (ODEs) that describe the time evolution of infection (virus as a function of time) with the implicit assumption that all cells see all virus and vice-versa, as discussed in several reviews [1–3]. A different approach from ODE MMs is to use agent-based MMs which treat the dynamics of each cell, and even each virus, individually and track local cell-virus interactions [4–6]. Such MMs are computationally intensive, restricting the number of cells, and therefore the area, which can be represented, and often lack support or validation in the form of experimental localization data [5, 7, 8]. Infection localization can also be modelled by dividing the human respiratory tract (HRT) into compartments corresponding to different regions of the HRT, where each compartment has different parameters based upon physiological differences [9, 10].

IAV infection spread faces a tremendous physical, spatial barrier in the form of the thick mucus layer which lines the HRT [11, 12]. This mucus layer sits atop the PCF, which itself sits atop and bathes the epithelial cells that line the HRT. As the ciliated epithelial cells' coordinated, beating cilia push the mucus layer upwards, it also pushes the PCF and the virus it contains, upwards towards the nose and mouth, and out of the HRT [13, 14]. Virus entering via the airways above the mucus layer, or progeny virus released at the base of the PCF by infected cells, can either become trapped by the mucus acting like flypaper, or remain in the PCF and get carried along with it, upwards and out of the HRT. These effective clearing mechanisms, jointly referred to as the mucociliary escalator, are typically taken somewhat into account in non-spatial ODE MMs via a term for the exponential clearance of virions. This simplification, however, does not account for the fact that not all cells will have equal exposure to the virus: cells that are downstream of the PCF flow (higher in the HRT) will have greater exposure to virus than those upstream, located deeper within the HRT. To our knowledge, the effect of virus entrainment due to the upwards advection of the PCF on IAV infection localization and spread within the HRT has never been evaluated.

In this work, a spatiotemporal MM for the spread of IAV infection within the HRT using partial differential equations (PDEs) is constructed. Through its one-dimensional representation of the HRT, the MM is used to study the effect of viral transport modes on the course of an IAV infection in vivo. Via the addition of two spatial parameters whose value is relatively well-established, namely the rate of diffusion and advection of virions, the MM produces a richer range of IAV kinetics, and predicts spatial infection spread patterns consistent with that observed in mice [15, 16]. The further addition of cellular regeneration and a simplified immune response allow the MM to reproduce levels of tissue damage consistent with that reported in patients [17]. The PDE MM is also used to explore differences in the kinetics of IAV infection, in the absence and presence of antiviral therapy, for IAV infection with either a seasonal or an avian-adapted strain, with the latter being typically more severe [18].

## Results

### A spatial MM for IAV in the HRT

In the proposed spatial MM, the HRT is represented as a one-dimensional tract running along the $x$-axis indicating the depth within the HRT, with $x = 0$ cm located at the top of the HRT (nose), and $x = 30$ cm terminating somewhere within the bronchi [19, 20], as illustrated in Fig 1. It is an extension of the 'standard' MM for IAV in vitro [21–23] which adds: (1) the diffusion of virions through the periciliary fluid (PCF) which lies between the cells' apical surface and the thick mucus blanket which lines the airways; (2) their advection due to the PCF being pushed along by the ciliated cells' beating cilia; and (3) their effect on the one-dimensional, depth-dependent fraction of non-motile (stationary) cells in various stages of infection. The spatial MM is formulated as

$$
\begin{aligned}
\frac{\partial T(x,t)}{\partial t} &= -\beta T(x,t)V(x,t) \\
\frac{\partial E_1(x,t)}{\partial t} &= \beta T(x,t)V(x,t) - \frac{n_E}{\tau_E}E_1(x,t) \\
\frac{\partial E_i(x,t)}{\partial t} &= \frac{n_E}{\tau_E}E_{i-1}(x,t) - \frac{n_E}{\tau_E}E_i(x,t) \qquad i = 2,3,...,n_E \\
\frac{\partial I_1(x,t)}{\partial t} &= \frac{n_E}{\tau_E}E_{n_E}(x,t) - \frac{n_I}{\tau_I}I_1(x,t) \\
\frac{\partial I_j(x,t)}{\partial t} &= \frac{n_I}{\tau_I}I_{j-1}(x,t) - \frac{n_I}{\tau_I}I_j(x,t) \qquad j = 2,3,...,n_I \\
\frac{\partial V(x,t)}{\partial t} &= p\sum_{j=1}^{n_I}I_j(x,t) - cV(x,t) + D_{\text{PCF}}\frac{\partial^2 V(x,t)}{\partial x^2} + v_a\frac{\partial V(x,t)}{\partial x}
\end{aligned}
\tag{1}
$$

The fraction of uninfected, target cells, $T(x,t)$, located at a depth $x$ at time $t$ are infected at rate $\beta V(x,t)$, proportional to the concentration of virions, $V(x,t)$, at that depth and time.

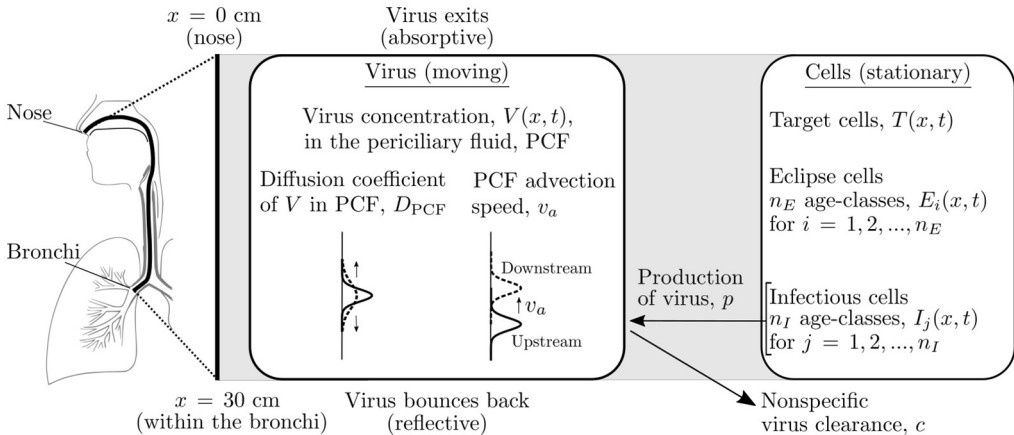

**Fig 1. Representation of the human respiratory tract by the mathematical model.** The MM Eq (1) represents the HRT as a one-dimensional track, as illustrated. The MM considers only the virus that is located in, and diffusing within the PCF, which also moves along with the PCF upwards at a fixed advection speed. What happens to the virus when it reaches the bottom (reflective) or top (absorptive) of the MM-represented HRT is indicated. MM (1) describes the interactions between the moving virus concentration in the PCF, and the stationary cells that carpet the HRT.

Newly infected cells are in the eclipse phase, $E_i(x, t)$, i.e. they are infected but are not yet producing virus. After an average time $\tau_E$, infected cells leave the eclipse phase to become infectious cells, $I_j(x, t)$, producing virions at constant rate $p$ for an average time $\tau_I$ until they cease virus production and undergo apoptosis. As in [21, 22], the eclipse and infectious phases are each divided into $n_E = n_I = 60$ age classes so that the time spent by cells in each phase follows a normal-like distribution, consistent with biological observations [24].

The MM assumes virions are released from stationary infected cells into the PCF, so $V(x, t)$ is the concentration of virus in the PCF. The produced virions diffuse through the PCF at rate $D_{PCF}$, and are transported upwards (towards the nose) with the PCF at speed $v_a$, as the PCF is pushed along by the beating cilia of the ciliated cells that line the HRT [13, 14]. The diffusion coefficient of IAV in the PCF, $D_{PCF}$, is estimated based on the Stokes-Einstein equation for IAV diffusing in plasma at body temperature, namely $D_{PCF} \approx 10^{-12}$ m$^2$/s [25, 26]. The advection speed of the PCF is set to that of the mucus layer, $v_a = 40$ μm/s, based on experiments by Matsui et al. [27] wherein microspheres 0.2 μm in diameter, located within the mucus and the PCF, in human tracheobronchial epithelial cell cultures grown in air-liquid interface, were found to travel at the same speed. At the edges of the mathematically modelled HRT, when virus in the PCF reaches the top of the HRT, it is lost (absorptive boundary conditions), and its upwards advection ensures it cannot reach the bottom of the modelled HRT, which becomes irrelevant, as will be shown later. Absorption of virions into the mucus blanket (out of the PCF), their loss of viral infectivity over time (in the PCF), and other modes of non-specific virion clearance are all taken into account via a single exponential viral clearance rate term, $cV(x, t)$. Virions contributing to the infection in this MM are those remaining in the PCF, in direct contact with infectable cells.

In the spatial MM simulator, infection is initiated by a spatially localized, spray-like virus inoculum deposited at depth $x_d$ within the HRT. The spray-like inoculum is represented by a Gaussian centred at the site of deposition, with a standard deviation of 0.5 mm, about 10× the size of a large cough droplet [28]. At sites far from $x = x_d$, $V(x, t = 0) \approx 0$. The baseline values of the spatial MM's parameters were all set to values estimated in Baccam et al. [29], obtained by fitting a non-spatial MM to patient data from experimental primary infections with the influenza A/Hong Kong/123/77 (H1N1) virus. Table 1 lists the initial conditions and parameters used. A complete description of the spatial MM is provided in the Methods.

## IAV kinetics in the presence of virus diffusion and advection

Infection of cells by the IAV, and the subsequent release of virions, occurs almost exclusively at the apical surface of the epithelium [30, 31]. When advective motion of the PCF is neglected, virions can be assumed to undergo Brownian motion and the virus concentration distribution in the PCF will be governed by diffusion. Fig 2(a)–2(c) shows IAV infection kinetics in the presence of diffusion alone, where quantities are shown averaged over all spatial sites as a function of time (see Methods). The infection takes place at a much slower pace in the spatial (diffusion only) MM than in the non-spatial MM which implicitly assumes an infinite diffusion coefficient. Fig 2(d)–2(f) shows the spatial extent of the infection dissemination (with diffusion only) through the HRT at certain times post-infection. Virus becomes available to target cells gradually as it diffuses away from the site of initial infection, $x_d = 15$ cm, and the infection wavefront moves outwards from that site symmetrically towards the two ends of the HRT at $x = 0$ and $x_{max}$. S1 Video depicts the spatiotemporal course of the infection, in the presence of diffusion only, in the form of a video.

There is an important distinction to be made between the *actual* virus concentration in the PCF and the virus concentration *measured experimentally*. In a typical, non-spatial MM, the

**Table 1. Default initial conditions and parameter values.**

| Symbol | Description | Value* [Source] |
|---|---|---|
| | — Spatial simulator parameters — | |
| $D_{PCF}$ | IAV diffusion coefficient in PCF at 37°C | $10^{-12}$ m$^2$/s [25] |
| $v_a$ | advection speed of PCF | 40 μm/s [27] |
| $\Delta x$ | size of one spatial grid site | 100 μm ($N_x$ = 3000 sites) [see Methods] |
| $\Delta t$ | duration of one time step | 2.5 s ($\Delta t = \Delta x/v_a$) [see Methods] |
| | — Initial conditions — | |
| $x_d$ | deposition depth of virus inoculum | 15 cm |
| $\langle V(x, t = 0)\rangle$ | initial virus inoculum averaged over $x$ | $7.5 \times 10^{-2}$ TCID$_{50}$/mL [29, *] |
| $T(x, t = 0)$ | initial fraction of uninfected target cells | $1.0 \, \forall \, x$ |
| $E_i, I_j(x, t = 0)$ | initial fraction of infected cells | $0.0 \, \forall \, x$ |
| | — Infection parameters — | |
| $\tau_E$ | duration of eclipse phase | 8 h ($n_E = 60$) [*] |
| $\tau_I$ | productively infected cell lifespan | 20 h ($n_I = 60$) [*] |
| $c$ | virus clearance rate | 0.22 h$^{-1}$ [29] |
| $\beta$ | infection rate of cells by virus | $1.33 \times 10^{-6}$ (TCID$_{50}$/mL)$^{-1}$ · h$^{-1}$ [29] |
| $p$ | virus production rate | $8.4 \times 10^{-6}$ (TCID$_{50}$/mL) · h$^{-1}$ ($11 \times p_{Baccam}$) [*] |

* These values are used in all simulations unless otherwise stated. The spatially averaged initial inoculum, $\langle V(x, t = 0)\rangle$, equals the value in [29] (see Methods). Values for $\tau_E$ and $\tau_I$ were chosen so as to lie near the middle of the ranges of [6, 10] h and [10, 40] h, respectively, obtained from MMs of IAV infections in vitro [21, 22]. The value for $p$ is discussed in the next Section, and that for $p_{Baccam}$ is explained in Methods.

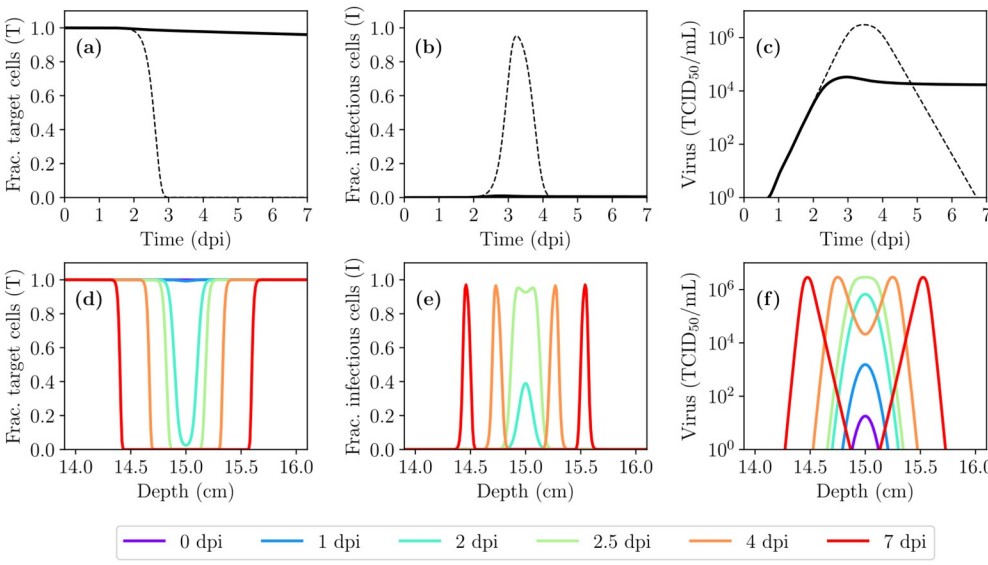

**Fig 2. IAV infection kinetics in the presence of diffusion alone (no advection).** (Top) Time course (averaged over space) of the infection for the fraction of cells in the (a) target/uninfected or (b) infectious state, and (c) the infectious virus concentration, obtained using the non-spatial ODE MM (dashed) or the spatial (diffusion only) MM (solid). (Bottom) Localized fraction of cells in the (d) target or (e) infectious state and (f) the infectious virus concentration as a function of depth within the HRT shown at specific days post-infection (dpi).

virus concentration in the PCF is implicitly assumed equal everywhere, $V(t)$, and can be compared directly to the experimentally measured concentration at that time $t$. In the spatial MM, the virus concentration depends not only on time but also on depth ($x$), i.e. $V(x, t)$ rather than just $V(t)$. The virus concentration curves shown herein when the $x$-axis is time rather than depth, correspond to the concentration averaged over all values of $x \in [0, 30]$ cm. S1 Fig explores the impact of this choice on our findings below.

Along the length of the HRT, a layer of mucus about 0.5 μm–5 μm thick covers the PCF [11]. The collective motion of the underlying epithelial cells' beating cilia, dubbed the mucociliary escalator, drives this mucus layer upwards. It also leads to an upward advection of the PCF, at a speed similar to that of the mucus layer [27, 32], entraining any virus in the PCF upwards at that speed. Given the advection speed of the mucus and PCF ($v_a \approx 40$ μm/s), any newly produced or deposited virion would be cleared from the HRT in less than $\sim 12$ min (30 cm/$v_a$). Thus, it is not surprising that adding advection to the spatial MM, while still using the same parameter values, results in a subdued, low viral titer, slow-growing infection which still has not peaked by 7 dpi. In contrast, viral titer in patients infected with influenza A/Hong Kong/123/77 (H1N1) virus in Baccam et al. [29] peaks 2–3 dpi, which the non-spatial MM with these same parameters reproduces well.

Fig 3(a)–3(c) shows the IAV infection kinetics in the presence of both diffusion and upward advection as the virus production rate, $p$, is increased from the base value estimated by Baccam et al. [29] for their non-spatial MM, $p_{\text{Baccam}}$ (see Methods). Increasing the virus production rate to $11 \times p_{\text{Baccam}}$ yields an infection that peaks at $\sim 3$ dpi. This adjusted value for the virus production rate ($p = 11 \times p_{\text{Baccam}}$, see Table 1) is used in the remainder of this work, unless stated otherwise, so that the spatial MM qualitatively reproduces the approximate timing of viral titer peak in an IAV infection.

Using the new value for the virus production rate, Fig 3(d)–3(f) shows the spatial extent of the infection dissemination in the presence of both diffusion and advection in the spatial MM. It illustrates the protective effect of advection: preventing the infection from travelling much beyond its initial deposition depth ($x_d = 15$ cm), as seen from the target cell depletion shown in Fig 3(d). Fig 3(g)–3(i) explores the effect of varying this depth of deposition of the initial virus inoculum, $x_d$. When the inoculum deposits lower in the HRT (as $x_d$ increases), the fraction of HRT consumed by the infection increases, resulting in higher viral titer peak and total virus yield. While such differences in viral titer could have clinical implications, the corresponding changes in viral titer would not be experimentally detectable in light of typical uncertainties (at least 10-fold) in titer measurements from nasal washes.

In the presence of both diffusion and advection in the spatial MM, the initial virus inoculum travels quickly from its deposition site at a fixed speed up the HRT, leaving in its wake a small, equal fraction of infected cells at all sites above the deposition point ($\forall x < x_d$). As the newly infected cells begin to release virus which is entrained up the HRT, uninfected cells are only exposed to virus produced by cells lower than (upstream of) their own depth (greater $x$ value), such that cells higher in the HRT (smaller $x$ value) are exposed to the most virus. Consequently, infection in the upper HRT proceeds much faster than that in the lower HRT. Fig 3(j)–3(l) shows the localized fraction of target and infectious cells, and infectious virus concentration over time at specific sites or depths along the HRT. Despite infection kinetic parameters being the same at all sites, infection progresses faster at sites located higher in the HRT, downstream of the virus flow.

Interestingly, the more rapid progression of the infection higher in the HRT makes it appear as though the infection is moving downwards, from the top (nose, $x = 0$) towards the bottom ($x = 30$ cm) of the HRT. Both the target cell depletion wavefront in Fig 3(d) and the infected cell "pulse" in Fig 3(e) appear to be moving to the right (towards the lower HRT) as

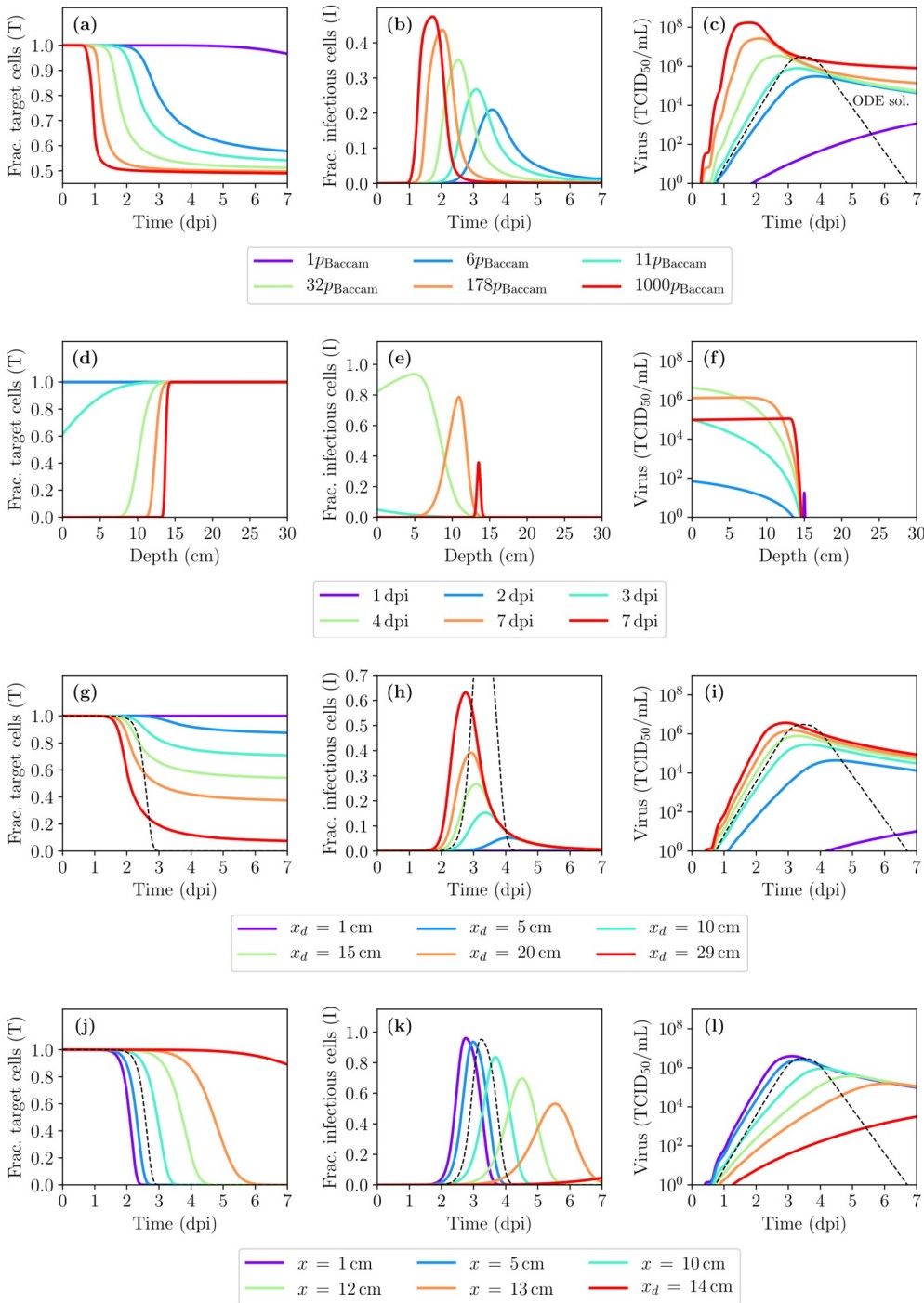

**Fig 3. IAV infection kinetics in the presence of diffusion and advection.** (a,b,c) Time course (averaged over space) of the infection for the fraction of cells in the (a) target/uninfected or (b) infectious state, and (c) the infectious virus concentration, obtained using the ODE MM (dashed) or the spatial MM (solid), as the rate of virus production, $p$, is varied. (d,e,f) Localized fraction of cells in the (d) target or (e) infectious state and (f) the infectious virus concentration as a function of depth within the HRT shown at specific days post-infection (dpi). (g,h,i) Same as (a,b,c) but varying the depth of deposition of the initial virus inoculum ($x_d$). (j,k,l) Similar to (a,b,c) but rather than being averaged over space, the infection time course is shown at specific, spatially localized depths ($x$). Unless otherwise noted, $p = 11 \times p_{\text{Baccam}}$ and $x_d = 15$ cm.

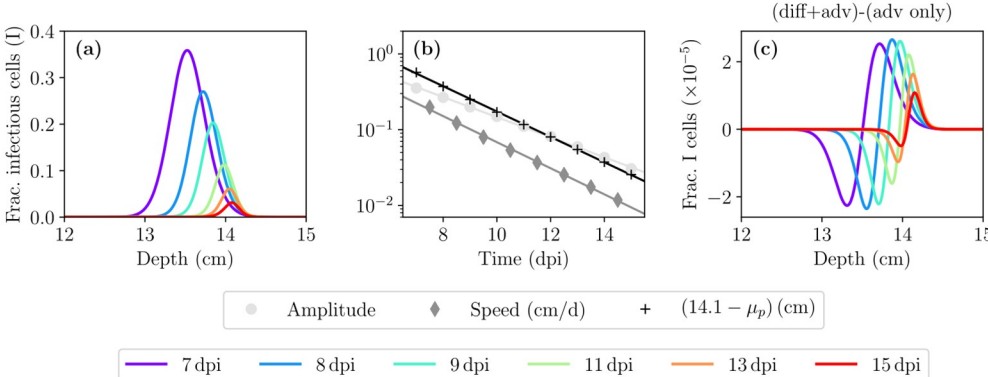

**Fig 4. A closer look at infection resolution: the impact of deposition depth and diffusion.** (a) A zoomed-in continuation of Fig 3(e) showing the localized fraction of infectious cells as a function of depth within the HRT, at times beyond 7 dpi. (b) The (nearly exponential) decay over time of the amplitude and speed of the Gaussian pulse of fraction of infected cells shown in (a), and of its centre location (mean, $\mu_p$) relative to its final, asymptotic location of 14.1 cm. The points are from the MM-simulated infection, and the lines are an exponential regression (linear regression to the log of the quantities). (c) The fraction of infectious cells shown in (a) in the presence of diffusion and advection (diff+adv) minus that obtained in the presence of advection only (adv only), i.e. when $D_{\mathrm{PCF}} = 0$.

time advances. This can be seen more clearly in the videos (S2 and S3 Videos) which depict the spatiotemporal course of the infection, in the presence of both diffusion and advection. Cells in the upper HRT are consumed first, after which the pulse of infected cells appears to slowly move backwards, down the HRT, back to the initial deposition site, where it then slowly dissipates. The observation that the infection appears to be progressing from the top towards the bottom of the HRT has also been noted in experimental studies of mice infected with IAV engineered to be fluorescent or bioluminescent [15, 16].

Fig 4(a) provides a more "zoomed-in" view of what happens to the pulse in Fig 3(e) at times beyond 7 dpi. The pulse is shrinking and slowing down as it approaches the initial inoculum deposition depth of $x_d = 15$ cm. Fig 4(b) quantitatively displays the (nearly exponential) decay of both the amplitude and speed of the pulse, as its peak's position ($\mu_p$) approaches its final, asymptotic position of $\sim 14.1$ cm, a little short of its initial deposition depth. Fig 4(c) shows the difference between the pulses obtained in the presence of both diffusion and advection minus that in the presence of advection alone (no diffusion). While advection dominates over diffusion, the latter still plays a role in allowing a small amount of virus to travel upstream, against the advection, which results in slightly more cells infected upstream and slightly less downstream. The effect of diffusion in the presence of advection is most likely negligible (one part in $10^5$ in the fraction of cells infected) in light of typical, in vivo experimental variability ($\sim 10$-fold variability). As such, it could be omitted in future modelling efforts.

Finally, the robustness of these results to variations in the advection speed was evaluated. Fig 5(a)–5(c) shows the effect of the advection speed on the infection kinetics when the virus production rate is fixed to its value in the non-spatial model, $p_{\mathrm{Baccam}}$. In Fig 5(d)–5(f), the virus production rate is increased as advection speed is increased so that the spatial MM approximately reproduces the timing of viral titer peak in the non-spatial MM for all values of the advection speed. For the lowest advection speed considered ($v_a = 4$ μm/s), no adjustment to the virus production rate was needed. Fig 5(g)–5(i) shows the spatial extent of the infection at specific days post-infection for advection speed $v_a = 4$ μm/s and virus production rate $p_{\mathrm{Baccam}}$. Even at a speed $10\times$ slower than that measured experimentally [27], the MM predicts advection would still prevent the infection from travelling beyond its initial deposition depth (herein

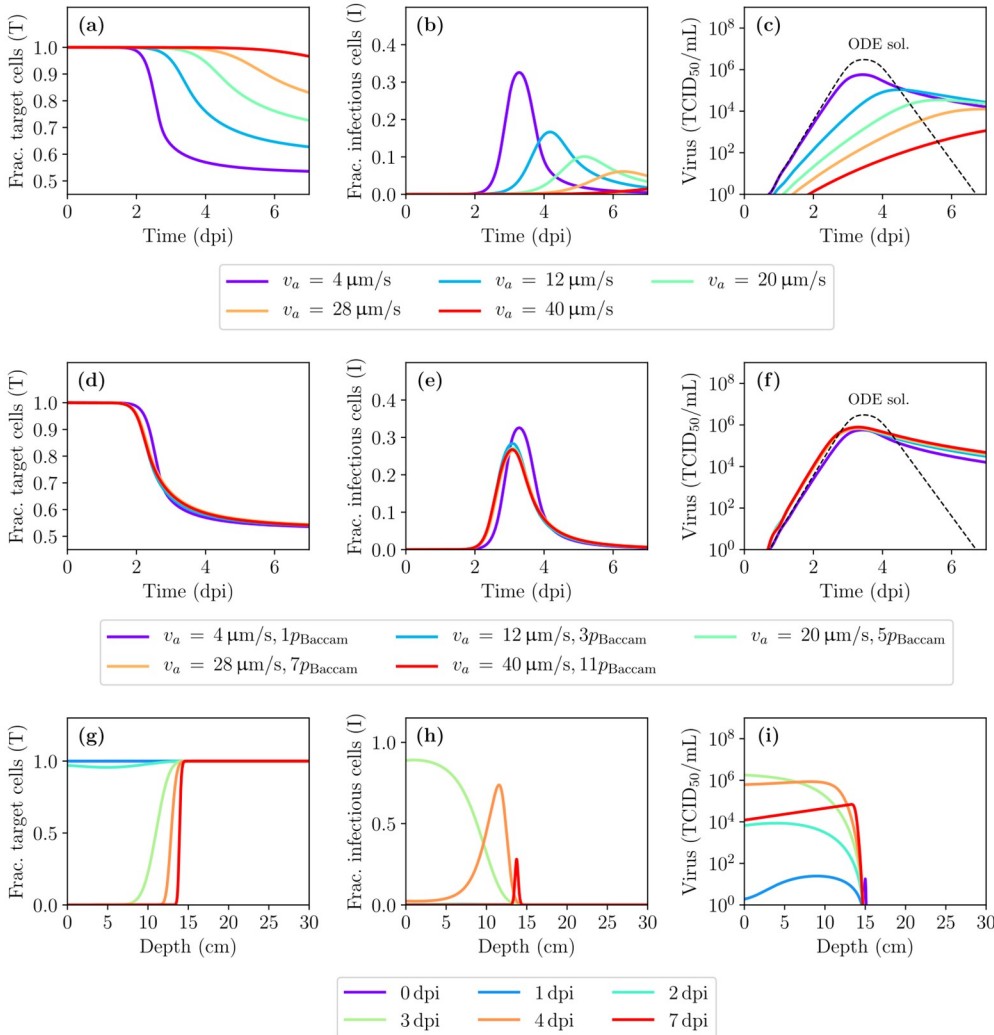

**Fig 5. The effect of advection speed on the MM predictions of IAV infection kinetics.** (a–f) Time course (averaged over space) of the infection for the fraction of cells in the (a,d) target/uninfected or (b,e) infectious states, and (c,f) the infectious virus concentration, as the advections speed ($v_a$) is varied either (a,b,c) on its own; or (d,e,f) along with the virus production rate so as to keep virus peak titer and timing constant. (g–i) Spatial distribution of the infection at various days post-infection for an advection speed $v_a$ = 4 μm/s and virus production rate $p_{Baccam}$.

$x_d$ = 15), and the infection still appears to be progressing downwards as the infection peaks and resolves faster in the upper than in the lower HRT.

## Target cell regeneration during IAV infection

During an uncomplicated IAV infection, viral loads typically fall below detectable levels between 6 dpi–8 dpi [33, 34]. An experimental study in which mice were sacrificed at various times over the course of an IAV infection [35] found that by 3 dpi, ciliated cells were rarely observed on the tracheal surface, and instead the latter was mostly composed of a layer of undifferentiated cells. By 5 dpi, signs of repair were apparent, by 10 dpi the tracheal surface was covered with ciliated and nonciliated cells, and by 14 dpi the surface was indistinguishable from the uninfected surface. Another experimental study in hamsters suggests that 6 d–7 d after mechanical injury, most of the epithelium is comparable to that of control hamsters [36].

A similar study in guinea-pigs suggests that 5 d after mechanical injury, the epithelium was ciliated and differentiated, and by 15 d it was indistinguishable from that of control guinea-pigs [37]. Together, these studies suggest that the process of cellular regeneration following damage caused by an IAV infection would at least begin, if not be well underway, before the IAV infection has fully resolved.

Following airway injury, numerous factors are thought to trigger the start of the repair process [38]. Once triggered, it is believed neighbouring epithelial cells will try to stretch to cover the denuded area, and divide so as to regain their normal shape while maintaining coverage. As damage becomes more significant, progenitor cells, mainly basal cells exposed to the PCF as the epithelial cells above them detach or are removed by apoptosis or damage, also undergo proliferation and differentiation until the epithelium is restored to its pre-injury state [39]. With these processes in mind, cellular regeneration was implemented in the spatial MM by replacing the equation for uninfected, target cells in MM Eq (1) with

$$\frac{\partial T(x,t)}{\partial t} = -\beta T(x,t)V(x,t) + r_D T(x,t)D(x, t - \tau_D) \ , \tag{2}$$

where

$$D(x,t) = 1 - \left[ T(x,t) - \sum_{i=1}^{n_E} E_i(x,t) - \sum_{j=1}^{n_I} I_j(x,t) \right]$$

is the fraction of dead cells, i.e. the extent of the damage or injury. As such, cellular regeneration in Eq (2) proceeds at a rate that is greater in the presence of greater damage (higher $D$) and of greater fraction of cells available to regenerate (higher $T$). Parameter $r_D$ sets the scale of the regeneration rate (which also depends on $T$ and $D$), and $\tau_D$ is the regeneration delay such that the current regeneration rate depends on the fraction of target cells ($T$) currently available to repopulate the area, and on the amount of damage ($D$) that was perceived some time $\tau_D$ ago. The delay between damage and regeneration accounts for the time required for the damage to activate appropriate signalling pathways, and for both division and differentiation to take place so that newly regenerated cells are susceptible to infection. This same equation has been used by others to represent target cell regeneration during an IAV infection [40, 41], but the authors therein did not include a delay ($\tau_D = 0$).

An experimental study of cell regeneration in hamsters [36] suggests that in many cases, between 18 h–24 h following a mechanical injury, cells have already stretched or migrated to cover the complete denuded area and cell division has begun. A similar study in guinea-pigs [37] also suggests that at 15 h after mechanical injury, cells have migrated to cover the denuded area and cell proliferation is underway. This suggests that the delay between damage and the start of cellular division is around 15 h–24 h. Herein, this delay—namely the time elapsed between damage and the start of both cellular division and differentiation to an extent that newly regenerated cells are susceptible to infection—was chosen to be $\tau_D = 1d$, based on these experimental studies. To select an appropriate value for the regeneration rate ($r_D$), HRT epithelial cell regeneration following mechanical injury was simulated using Eq (2). Results are shown in Fig 6 for different regeneration rates and delays: the regeneration rate determines the steepness of the regeneration, while the regeneration delay sets its timing. These parameters should be easily identifiable if experimental data was available. An intermediate value of $r_D = 0.75$ d$^{-1}$ was chosen to ensure that with a delay of 1 d, regeneration is well underway by 5 d–8 d, and completely resolved by 12 d–14 d [35–37].

Fig 7 shows the infection kinetics in the presence of cellular regeneration. At this stage, and over the range of values considered here, the spatial MM is insensitive to the choice of

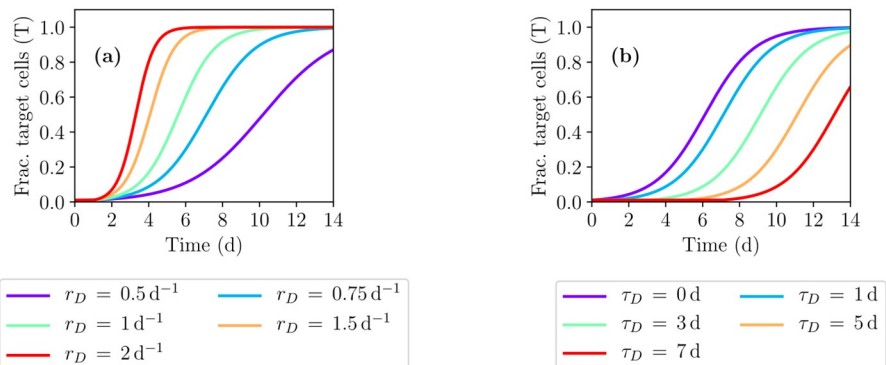

**Fig 6. Target cell regeneration following a MM-simulated mechanical injury.** Regeneration from mechanical injury was simulated using Eq (2) in the absence of infection, i.e. $V(x, t) = E_i(x, t) = I_j(x, t) = 0$ and $D(x, t) = 1 − T(x, t)$. Initially, $T(x, t = 0) = 0.01$, $D(x, t = 0) = 0.99$, and $D(x, t < 0) = 0$, representing an injury inflicted at $t = 0$ which removed 99% of all target cells. The effect of varying (a) the regeneration rate ($r_D$) or (b) the regeneration delay ($\tau_D$) are shown. Unless varied, $\tau_D = 1$ d and $r_D = 0.75$ d$^{-1}$.

regeneration rate or its delay. This is largely due to the fact that, in the absence of an immune response, the HRT above the inoculum deposition point is decimated by the infection at a rate much faster than the density-dependent regeneration can counter. As damage increases, the number of target cells available to replenish the lost cells also decreases dramatically, preventing any significant regeneration. To get a more realistic view of the role and impact of cellular regeneration on infection kinetics, the protective role of the immune response must be considered.

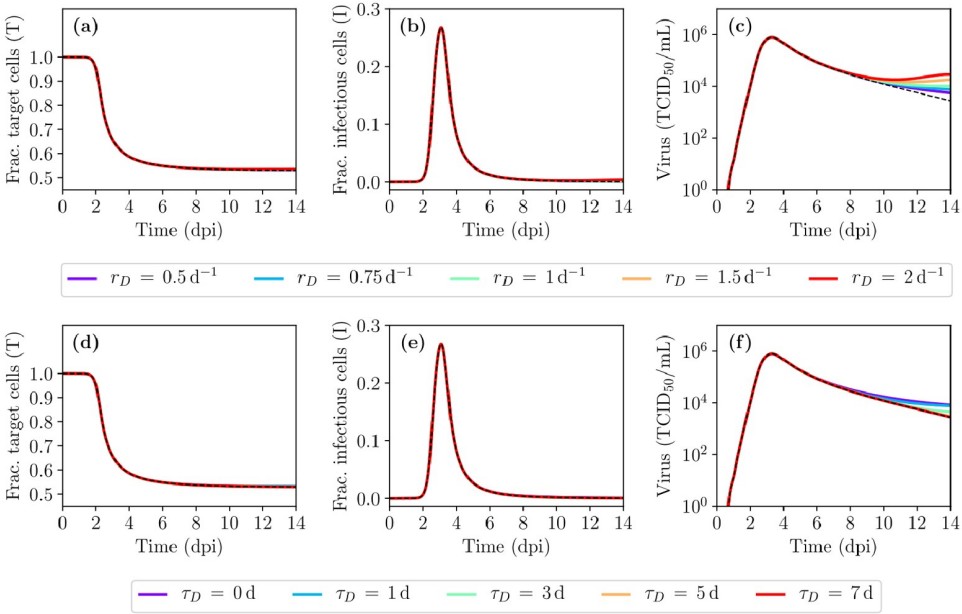

**Fig 7. The effect of cellular regeneration on the MM prediction of IAV infection kinetics.** The effect of varying (a–c) the regeneration rate ($r_D$) or (d–f) the regeneration delay ($\tau_D$) are shown. Unless varied, $\tau_D = 1$ s and $r_D = 0.75$ d$^{-1}$.

### Immune response to an IAV infection

Past MM for IAV infection have incorporated one or more immune response components with varying degrees of success [1, 41, 42]. Difficulties in incorporating an immune response when modelling IAV infections include the complex nature of the interactions (large networks of cells and signals), and the lack of appropriate data (quantity and quality) to inform the MMs [3]. The simplified immune response considered herein comprises an innate response based on interferon (IFN), a humoral response represented by antibodies (Abs), and a cellular response embodied by cytotoxic T lymphocytes (CTLs). Since our aim is to display the MM's range of kinetics—rather than identify parameters, analyze data, or challenge hypotheses—a parsimonious approach was preferred wherein key immune components are represented using empirical curves that broadly reproduce the scale and timing of these experimentally measured quantities (see Methods). Fig 8 shows the empirical MM curves against their corresponding experimental time courses, for IFN, Abs, and CTLs, during experimental, in vivo IAV infections.

IFN is known to have many effects [49–51], including making cells resistant to infection and recruiting additional immune factors and cells to locally enhance cell killing and virus neutralization. For simplicity, its effect herein is to reduce the rate of virus production $p$ [29, 52], via resistance parameter, $f_{50}$, analogous to the $IC_{50}$ used to describe antiviral resistance. The resistance is defined such that if $f_{50} = 0.8$, the virus production rate is halved when IFN concentration is 80% of its peak value (see Methods). Fig 9(a)–9(c) shows the effect of IFN in the MM as resistance to IFN, $f_{50}$, is lowered (increased sensitivity). The initial viral titer peak is reduced due to IFN presence, with lower resistance (smaller $f_{50}$) having a greater impact. However, once IFN starts to decay, its effect rapidly dissipates, leading to a rise, or even a rebound, in the viral load. On its own, IFN does not lead to infection resolution in this MM.

The effect of Abs herein, like elsewhere [17, 40, 47, 52, 53], is to enhance infectious virus clearance such that $c$ becomes $c + k_A A(t)$, where $k_A$ represents the neutralization rate of infectious virus by Abs, $A(t)$. Fig 9(d)–9(f) shows the effect of IFN+Abs in the MM as the neutralization rate, $k_A$, is increased. Low binding affinity Abs ($k_A \leq 200 \text{ h}^{-1}$) cannot clear the infection. As $k_A$ increases, viral titer decay rates increase, leading to infection resolution within

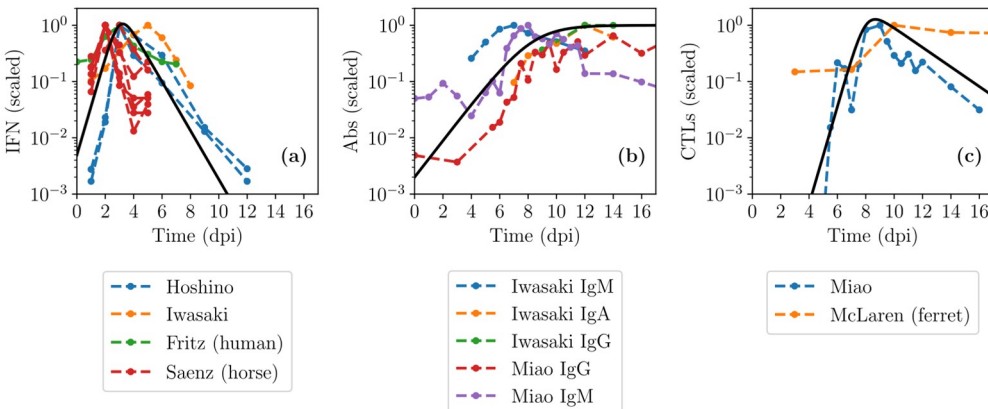

**Fig 8. Time courses of key immune response components during IAV infections in vivo.** Empirical MM curves (solid, black) are shown against experimental measurements (dashed, coloured) for (a) interferon (IFN) [43–46], (b) antibodies (Abs) [45, 47]; and (c) cytotoxic T lymphocytes (CTLs) [47, 48], taken over the course of in vivo IAV infections in mice, unless otherwise specified.

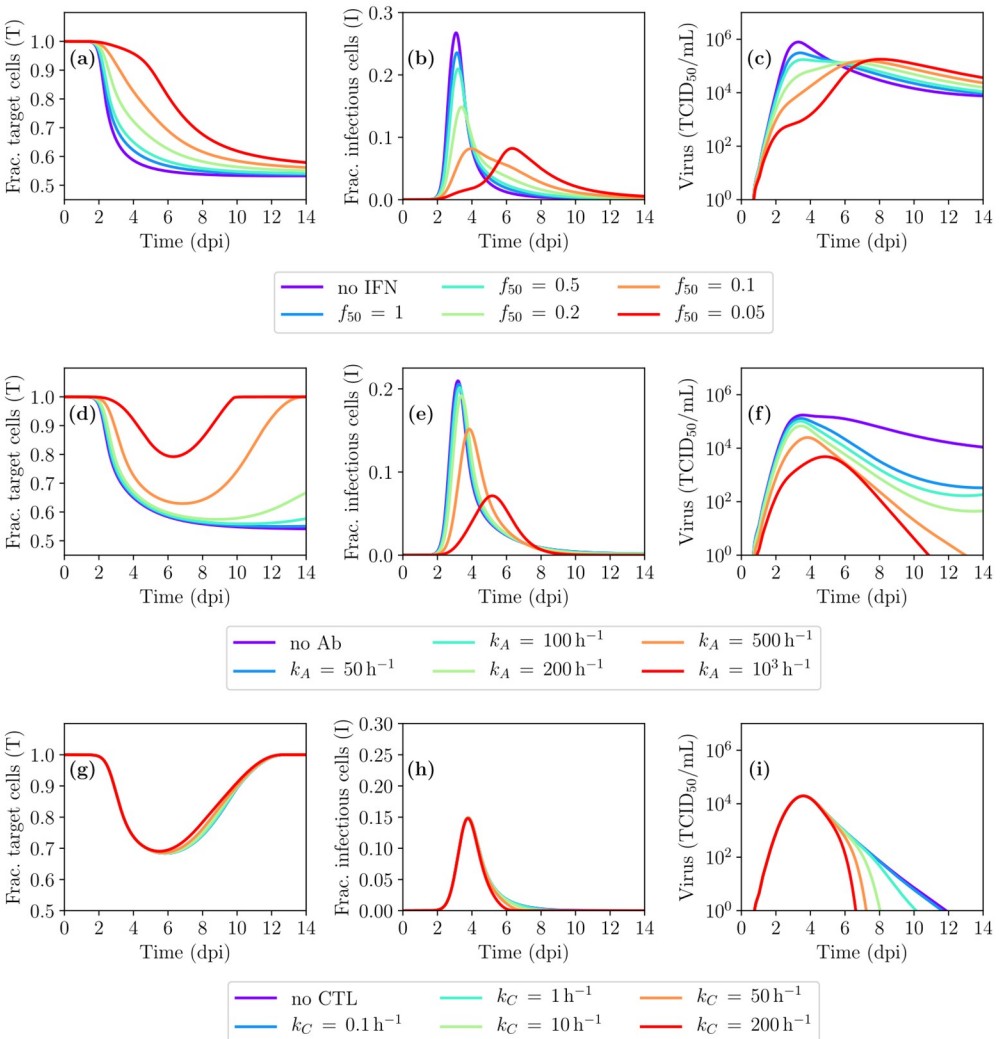

**Fig 9. Spatial MM-predicted IAV infection in the presence of key immune response components.** (a–c) The effect of interferon (IFN) as IFN resistance, $f_{50}$, is decreased (sensitivity is increased). (d–f) The combined effect of IFN and antibodies (Abs) as the rate of infectious virus neutralization by Abs, $k_A$, is increased. (g–i) The combined effect of IFN +Abs and cytotoxic T lymphocytes (CTLs) as the rate of infected cell killing by CTLs, $k_C$, is increased. Unless otherwise noted, $f_{50} = 0.5$ and $k_A = 500\,\mathrm{h}^{-1}$.

11dpi–16dpi. However, the time to infection resolution remains longer than the 6dpi–8dpi typically observed for IAV infections in humans [33, 34].

The effect of CTLs herein, like elsewhere [17, 40, 47, 53], is to increase the rate of loss of infected cells expressing IAV peptides on their MHC-1. Infected cells begin expressing IAV peptides ∼4 h post-infection [17, 54], or approximately halfway through the eclipse phase (8 h in the spatial MM, see Table 1), such that all infected cells past the mid-point of their eclipse phase will be removed at rate $k_C C(t)$, where $k_C$ represents the killing rate of infected cells by CTLs, $C(t)$. Fig 9(g)–9(i) shows the effect of IFN+Abs+CTLs in the MM as the killing rate, $k_C$, is increased. Higher CTL killing rates ($k_C$) cause the infection to resolve earlier.

Fig 10 shows the MM predictions and experimental data of immune response knockout experiments, wherein one component of the immune response is suppressed or neutralized.

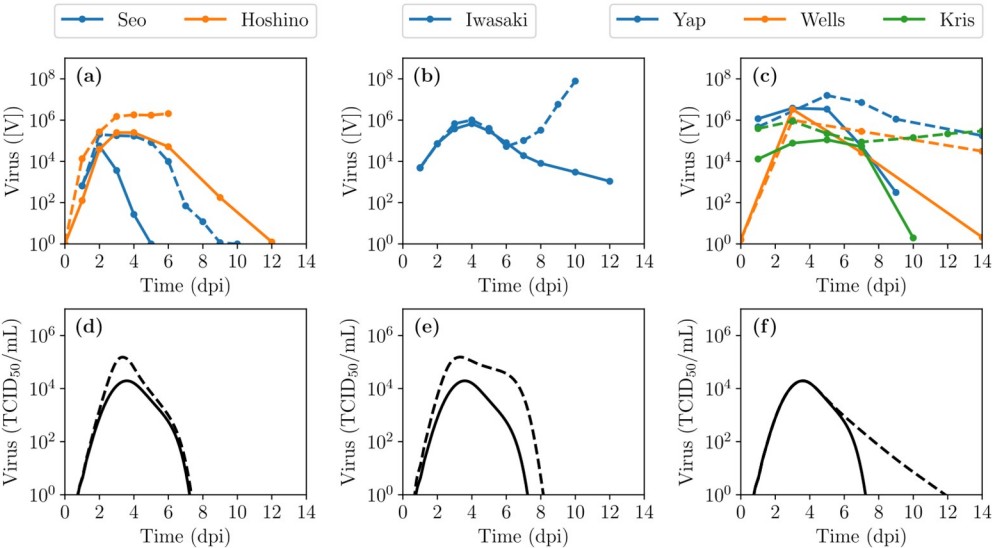

**Fig 10. MM predictions vs experimental data of immune response knockout experiments.** Experimental (top row) or MM-simulated (bottom row) viral titer time course for IAV infections with a full immune response (solid lines) or with one immune response component experimentally or mathematically disabled or knocked-out (dashed lines), respectively. Either (a,d) the IFN response [46, 55], (b,e) the Ab response [45] or (c,f) the CTL response [56–58] were disabled. In these experimental studies, the animal model is mice, except Seo et al. [55] which was conducted in pigs. Experimental viral load is measured in $TCID_{50}$/mL for [55], in $TCID_{50}$/mouse (of lung homogenate) for [46], in $EID_{50}$/mL for [56], [58] and for [57] and in pfu/mL for [45]. The MM results are shown with $f_{50} = 0.5$, $k_A = 500 \text{ h}^{-1}$ and $k_C = 50 \text{ h}^{-1}$, or disabled with $F(t) = 0$, $A(t) = 0$ or $C(t) = 0$ in (d,e,f), respectively.

The MM prediction of an IFN knockout experiment is in good agreement with the experimental data of IFN knockout experiments [46, 55] at early time as both suggest IFN acts early and helps to reduce the initial viral titer peak. The experimental Ab knockout experiment [45] suggests Abs help reduce viral load at a later time in the infection but does not affect the initial viral titer peak. The MM prediction also suggests Abs help reduce the viral load at a later time but also affects the initial viral titer peak. This could be better reproduced by the MM by having a lower initial amount of Abs ($A_0$) so that Abs appear later and thus act later. Finally, the MM prediction of the CTL knockout experiment is in good agreement with the experimental data of CTL knockout experiments [56–58] as both suggest that CTLs act only at a late stage in the infection, helping reduce infection duration. Generally, the experimental knockout appear to show a greater impact for the knock-outs than predicted by the MM. This is likely due to the intricate interaction network between the different immune response components in vivo which makes it difficult to experimentally disrupt one component without affecting another.

## Capturing the kinetics of in vivo infections with seasonal and avian IAV strains

The difference in infection severity between patients naturally infected with seasonal IAV strain or avian-origin H5N1 strains is thought to be the result of a number of different possible factors, including more rapid infection and replication kinetics, cell tropism, lack of pre-existing immunity, and/or an aberrant immune response [22, 55, 59–66]. Fig 11(a) shows viral load measurements from pharyngeal swabs of patients (each point is a single measure from a single patient) naturally infected with either a seasonal (H3N2 or H1N1) IAV strain or a strain of the

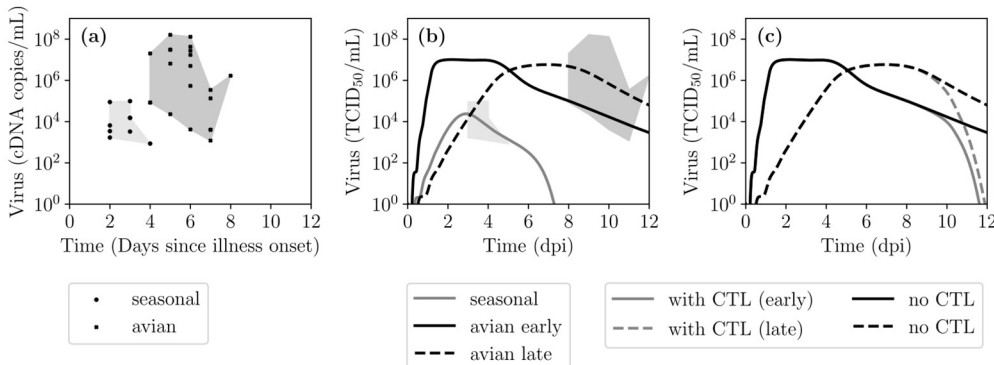

**Fig 11. Kinetics of infection in patients naturally infected with a seasonal or avian IAV strain.** (a) Total viral load measurements (cDNA/mL via qRT-PCR) from throat (pharyngeal) swabs of patients which naturally contracted infections with either a seasonal (H3N2 or H1N1) or avian (H5N1) IAV strain (data taken from [59]). The shaded polygons trace out the extent of the data points. (b) Proposed time courses for in vivo IAV infections with either a seasonal (grey) or H5N1 (black) IAV strain. The solid and dashed black lines represent two plausible time courses for infection with an IAV H5N1 strain: a more rapid rise to higher viral titers than for infections with seasonal strains (solid); or a more moderate rise, similar to that seen for infections with seasonal strains, that grows to higher titer due to lack of rapid and effective immune control (dashed). MM parameters for these curves are listed in Table 2. (c) The possible effect of a delayed CTL response, i.e. one which peaks at 12dpi with a killing efficacy of $k_C = 50$ h$^{-1}$ (grey) in both the early (solid) and late (dashed) infection time courses which are otherwise shown without a CTL response (black).

H5N1 subtype. This study wherein these measures were taken over the same time period and following the same methodology, such that viral loads for infection with seasonal and H5N1 strains can be readily compared, is the only one of its kind. Since patients were recruited into the study days after illness onset, and began receiving antiviral therapy immediately upon hospital admission, untreated infection kinetics leading up to and after admission are not available. The measured pharyngeal viral load (determined via qRT-PCR) is ~100-fold higher for patients infected with H5N1 than seasonal IAV strains. This could also be true of the infectious viral titer (typically measured via TCID$_{50}$ or PFU) which was not measured as part of this study, although the ratio of infectious to total IAV is known to change over the course of an infection, and inconsistently so between experiments [67]. Another feature is that H5N1-infected patients reported to the hospital 3 d–4 d later after illness onset than those infected with seasonal IAV strains. It is unclear whether this is due to a slower progression of infection with an H5N1 strain, or because patients infected with H5N1 strains mostly came from remote provinces and took longer to reach the city hospital than those infected with seasonal strains which came from the city itself or neighbouring provinces.

From the data in Fig 11(a), two hypothetical infection time courses or portraits were developed for infection with IAV H5N1 strains, and one for seasonal strains, shown in Fig 11(b). The IAV H5N1 portraits were constructed from the MM of the seasonal portrait by shifting parameters controlling cell-virus interactions and immunity, in keeping with differences believed to be responsible for that shift. These parameters are presented in Table 2. This approach was chosen due to the limited data available on IAV H5N1 infection, and is simply meant to display one of the MM's possible applications. It is not intended as proof that these differences in parameter values are responsible for the differences between a infection with seasonal or H5N1 IAV strain.

The portrait for infection with a seasonal strain peaks at ~10$^3$–10$^4$ TCID$_{50}$/mL at ~3dpi, and resolves by ~8dpi [68, 69]. For infection with a strain of the H5N1 subtype, two options

**Table 2. Parameter values used to reproduce the IAV infection kinetics in Fig 11.**

| IAV strain | $\beta([V]^{-1} \cdot d^{-1})$ | $p([V] \cdot d^{-1})$ | $\tau_E(h)$ | $\tau_I(h)$ | IFN, $f_{50}$ | Abs*, $A_0$ | CTLs, $k_C$ (h$^{-1}$) |
|---|---|---|---|---|---|---|---|
| seasonal | $1.0 \times 10^{-6}$ | $2.0 \times 10^{7}$ | 8 | 10 | 0.5 | $2.0 \times 10^{-3}$ | 50 |
| avian (early) | $1.5 \times 10^{-6}$ | $5.0 \times 10^{7}$ | 6 | 80 | 5.0 | $1.0 \times 10^{-5}$ | — |
| avian (late) | $1.0 \times 10^{-7}$ | $5.0 \times 10^{7}$ | 12 | 100 | 5.0 | $1.0 \times 10^{-5}$ | — |

* The rate of infectious virus neutralization by Abs was fixed to $k_A = 500$ h$^{-1}$ for all 3 infections.

are considered: the infection peaks early, at higher titers which are sustained over a longer period; or the infection grows at a rate similar to infection with a seasonal strain, but that growth is sustained over a longer period and thus peaks later, reaching higher titers than infection with a seasonal strain. For infection with an H5N1 strain, viral titer peak was chosen to be 100-fold higher, as observed for total viral loads, and the reported 3dpi –4dpi delay in hospital admission was captured either as a longer, sustained infection in the early time course, or as a longer delay to reach peak titer in the late time course. The shaded areas, which depict the extent of the data points in Fig 11(a) where time is based on days since illness onset, are shown time-shifted in Fig 11(b) where time is days post-infection. For a seasonal infection, the (pale grey) area was shifted to one day later since there is generally a delay of one day between infection and symptom onsets [33]. For avian infections, the (dark grey) area was shifted to 4 days later since reports suggest $\sim$3–5 days elapse between symptom onset and known potential exposure to infected poultry [70–72].

For cell-virus interactions, the early time course relies on increased virus production ($p$), increased virus infectivity ($\beta$), and a shorter eclipse phase ($\tau_E$), consistent with shifts in these parameters estimated from in vitro infections of A549 cells with seasonal (H1N1) versus H5N1 and H7N9 IAV strains [22]. In contrast, the late time course relies on lower virus infectivity and a longer eclipse phase, along with a higher virus production rate. The early and late portraits also depend on 8- and 10-fold longer periods of virus production (infectious cell lifespan, $\tau_I$), respectively, compared to that for cells infected with a seasonal strain. In capturing differences in immunity, both the early and late portraits of infection with IAV H5N1 strains rely on 10-fold increased resistance to the effect of IFN ($f_{50}$), decrease in pre-existing immunity in the form of a 200-fold decrease in the neutralizing effect of Abs, and the absence of a CTL response. The resistance of H5N1 strains to the effect of IFN-$\alpha$ and -$\gamma$ has been reported in vitro in porcine cell cultures and in vivo in pigs [55]. The decrease in the neutralizing efficacy of Abs is captured in the simplistic Ab MM utilized herein as a 200-fold decrease in their initial quantity ($A_0$) which means a longer time ($\sim$8 d longer) to reach maximal Ab activity, but can equally be captured by a 200-fold decrease in the neutralization efficacy ($k_A$) of Abs. A recruitment study of patients naturally infected with avian H7N9 IAV strains reported that the patient group with the earliest recovery, had a prominent CTL response by 10 d post admission and a prominent Ab response, 2–3 d later [73]. In contrast, the patient group with the most delayed recovery had an Ab and CTL response that remained low even after 20 d post admission. This suggests that an infection with an H5N1 strain, characterized by a prolonged viral shedding period, would have delayed Ab and CTL responses. Because the portraits in Fig 11(b) only go up to 12dpi, the effect of these delayed CTLs would not be apparent. Fig 11(c) illustrates what the time course of infection with an avian strain could look like in the presence of a delayed CTL response, but one which would be as effective (same $k_C$) as that for infection with a seasonal strain. It shows an important role for CTLs in controlling the persistent

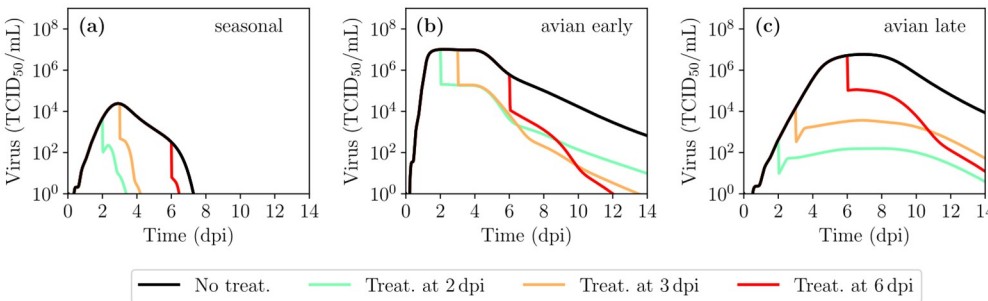

**Fig 12. MM-predicted NAI antiviral therapy efficacy in patients infected with a seasonal or avian IAV strain.** The viral titer time course for IAV infections under antiviral therapy initiated at various times post infection (see legend) with NAIs, captured as decreasing virus production rate. The effect of treatment is shown in the context of infection with (a) a seasonal IAV strain; or with an avian IAV strain based on either the (b) early or (c) late infection portraits. The drug efficacy, $\varepsilon$, was chosen to be 0.98. The other parameters used in the MM are those listed in Table 2.

shedding in these infections, consistent with the findings that a delayed CTL response correlated with slower recovery from infection with H7N9 strains [73].

Fig 12 explores the impact of antiviral therapy with neuraminidase inhibitors (NAIs), administered at various times post-infection, on the time courses for infection with seasonal or avian IAV strains. These results should be treated as hypotheses, as the infection time courses are hypothetical, and the results should serve as further display of the MM's possible future applications. Since NAIs block the release of newly produced virions, their effect is implemented in the MM, as elsewhere [42, 74–80], as a reduction in the virus production rate, namely $(1 - \varepsilon_{NAI})p$, from the time treatment is administered, where $\varepsilon_{NAI} \in [0, 1]$ is the drug efficacy. The efficacy of NAIs was set to $\varepsilon_{NAI} = 0.98$ to study the case of treatment with a high efficacy. Table 3 quantitatively compares the impact of NAI treatment for various endpoints: reduction in resolution time, peak viral titer, and area under the viral titer curve (AUC).

**Table 3. Measures of the impact of NAI treatment under various conditions** *.

| $t_{adm}$ (dpi) | $\Delta$[Resolution time (d)] | $\Delta$[AUC (d · TCID$_{50}$/mL)] | $\Delta$[Viral titer peak(TCID$_{50}$/mL)] |
|---|---|---|---|
| | | —Seasonal — | |
| 2 | $7.3 - 3.3 = 4.0$ | $10^{4.5}/10^{3.1} = 10^{1.4}$ | $10^{4.4}/10^{3.7} = 10^{0.7}$ |
| 3 | $7.3 - 4.2 = 3.1$ | $10^{4.5}/10^{4.2} = 10^{0.3}$ | $10^{4.4}/10^{4.4} = 10^{0.0}$ |
| 6 | $7.3 - 6.4 = 0.9$ | $10^{4.5}/10^{4.5} = 10^{0.0}$ | $10^{4.4}/10^{4.4} = 10^{0.0}$ |
| | | — Avian (early) — | |
| 2 | ND $- 17 =$ ND | $10^{7.5}/10^{6.9} = 10^{0.6}$ | $10^{7.0}/10^{7.0} = 10^{0.0}$ |
| 3 | ND $- 13 =$ ND | $10^{7.5}/10^{7.2} = 10^{0.3}$ | $10^{7.0}/10^{7.0} = 10^{0.0}$ |
| 6 | ND $- 12 =$ ND | $10^{7.5}/10^{7.5} = 10^{0.0}$ | $10^{7.0}/10^{7.0} = 10^{0.0}$ |
| | | — Avian (late) — | |
| 2 | ND $- 15 =$ ND | $10^{7.3}/10^{3.1} = 10^{4.2}$ | $10^{6.8}/10^{2.6} = 10^{4.2}$ |
| 3 | ND $- 17 =$ ND | $10^{7.3}/10^{4.4} = 10^{2.9}$ | $10^{6.8}/10^{4.2} = 10^{2.6}$ |
| 6 | ND $- 16 =$ ND | $10^{7.3}/10^{6.8} = 10^{0.5}$ | $10^{6.8}/10^{6.7} = 10^{0.1}$ |

* For NAI therapy administered at various times post infection ($t_{adm}$), the change ($\Delta$) in 3 commonly reported endpoints are shown for the viral titer curves in Fig 12. The change in resolution time is computed as a difference ($\Delta$ = without−with), whereas that for the area under the viral titer curve (AUC) and viral titer peak corresponds to the fold-change ($\Delta$ = without/with). ND stands for not determined in cases where infection resolution ($V(t) < 1$ TCID$_{50}$/mL) was not achieved without treatment.

Human volunteer studies in patients experimentally infected with seasonal (H1N1) IAV strains and treated with NAIs report that early treatment (24–32 h post infection) is effective in reducing viral load [81–83], and one such study reports that delayed treatment (50 h post infection) is less effective [82], in line with the MM results. A study of H5N1-infected patients recruited days after illness onset, and beginning NAI treatment immediately after admission, reports that late treatment (4–8 d after illness onset) is still effective in some patients, reducing viral load and thus possibly also reducing time to infection resolution. This would be consistent with some avian strain-infected patients experiencing an infection time course similar to the early portrait with moderate to no benefit from delayed NAI treatment, and others experiencing infection more consistent with the late portrait and benefiting from NAI therapy even when treatment initiation is delayed.

## Discussion

Mathematical models (MMs) for the course of an influenza A virus (IAV) infection in vivo typically assume the infection is spatially homogeneous, i.e. that all cells see all virus, and vice-versa, instantly over all space. This simplification is reasonable in vitro in cell culture infections whose spatial extent is often no more than one or two square centimetres [25], but it is unclear whether it remains appropriate at the scale of the entire HRT. With simplicity and parsimony in mind, the spatial MM introduced herein represents the HRT as a one-dimensional track that extends from the nose down to a depth of 30 cm. It implements two modes of viral transport: advection of virus upwards towards the nose, and diffusion of the virus within the periciliary fluid.

When diffusion alone is considered, infection in the spatial MM proceeds at a slower pace than in the non-spatial MM, but virus production is sustained for longer. The diffusion initially acts to spatially restrict the number of cells available to the infection, and then releases these cells progressively as the diffusing infection wave reaches ever further in the HRT. When advection is added to diffusion, the former dominates the infection kinetics, possibly requiring an increase in the virus production rate to restore the timing and level of viral titer peak to that in the non-spatial MM. This is noteworthy, firstly, because it shows that advection constitutes an effective physiological mechanism to suppress infection. Secondly, it shows that use of a non-spatial MM to analyze infection data possibly underestimates the virus production rate, and consequently also the total amount of virus produced over the course of an infection. Since the more virus are produced, the more mutations accumulate [74, 84, 85], underestimating the virus production rate means underestimating the rate and likelihood of emergence of drug resistance.

Interestingly, the MM suggests that the depth at which the initial virus inoculum deposits also plays a major role in determining the extent of HRT involvement in the infection. Specifically, the MM predicts no target cell consumption below the depth at which the initial virus inoculum deposits. While deposition at depths lower than $\sim 15$ cm did not result in experimentally measurable differences in the viral titer time course, initial depositions above that point both delayed and reduced peak titer. This is seen both for the basic, initial MM in Fig 3(g)–3(i), and in S2 Fig for the full spatial MM which includes cellular regeneration and an immune response. Whatever the true, biological deposition depth may be, in order to obtain equivalent viral titer time courses as one decreases the deposition depth (deposition at higher sites) in the spatial MM, one must increase the virus production rate to compensate for the fewer available target cells. This is one more way a non-spatial MM might underestimate the rate of virus production, and thus the likelihood of antiviral resistance emergence.

Herein, the depth of initial inoculum deposition in the MM was set arbitrarily to 15. In reality, the most likely site of deposition would be very difficult to determine. There are many possible routes for IAV transmission—e.g. inhalation of an expulsion spray from a sneeze or a

cough, direct contact with a contaminated person, or by indirect contact with a contaminated object—all of which are believed to play some role, with no consensus on the most likely or most common route [86]. Even if the route of transmission is known or assumed, for example from a cough, there is a large range of potential particle sizes for this mode of transmission [28], and particle size is believed to be inversely proportional to depth of deposition [87]. Even if the most likely depth of deposition was known, the site of initial particle deposition is unlikely to be the same as the site at which infection will first take hold because virus particles that deposit along the airways have to traverse the mucus and PCF layers to reach infection-susceptible epithelial cells, and advection in those layers would quickly drag virus away from its initial deposition site.

An unexpected finding was that although kinetically in the spatial MM the infection deposits at some depth and is dragged upwards by advection, the infection appears to be moving from the upper to the lower HRT. This is because the upper HRT is downstream of virus advection and sees most of the virus produced, while the lower HRT is upstream and sees little or none. Therefore, infection progresses, peaks and resolves faster in the upper HRT and slower in the lower HRT. It is this difference in time scale which makes it appear as though the infection is moving from the upper to the lower HRT. Experimental studies of mice infected with IAVs engineered to be fluorescent or bioluminescent observe infection spread from the upper towards the lower regions of the lung [15, 16], consistent with the MM's predictions. In the absence of sufficient immune control, this could serve to maintain the infection with, wherein the slower, lower HRT infection could re-ignite more rounds of infection in the upper HRT, depending on the delicate timing between infection time course in the lower HRT and cell regeneration in the upper HRT.

The spatial MM was expanded to include density-dependent cell regeneration, and a simplistic immune response comprising IFN acting to down-regulate the rate of virus production by infected cells, Abs neutralizing infectious virions, and CTLs killing infected cells. In the presence of this immune response, the MM-simulated IAV infection was well controlled and resolved fully by 8 dpi. The spatial MM could also largely reproduce the key features of in vivo immune response knockout experiments [1], although one should be cautious in extrapolating conclusions from murine experiments to humans, and in assuming that experimental disruption of one immune component can be carried out without affecting another. In the MM-simulated infection (see Fig 9g), 10% of the HRT was involved in the infection by 3 dpi, around 6 dpi the fraction of cells involved in the infection peaked at 30%, and by 10 dpi 10% of the epithelium had yet to regenerate, although viral titers had fallen below the detection limit (see Fig 9i). These numbers align well with a 1989 report in Russian cited by Bocharov and Romanyukha in their 1994 IAV MM paper [17] of 10% damage at symptom onset, 30–50% of the upper airway destroyed at the peak of the disease, and resolution of disease at a time when as much as 10% of the normal epithelium is still damaged.

The full spatial MM was applied to simulate the kinetics of infection with either a mild, seasonal IAV strain or a severe infection with an avian strain such as H5N1. Since complete, untreated, infection kinetic time courses for infection with avian IAV strains are not available, two different portraits were constructed to represent plausible time courses: one peaking early with sustained, high titers over several days, and another rising slowly, similarly to seasonal infections, but continuing on to reach high titers 3–4 days later. Differences in the hypothetical time courses for infection with a seasonal vs avian IAV strain could be captured by the MM by shifting the parameters in ways that are consistent with known or expected differences between infections with seasonal and avian strains. The portraits were used to evaluate possible impact of antiviral therapy with neuraminidase inhibitors. Treatment was most effective for the seasonal and late peak avian IAV strain portraits when administered early during the

infection at 2 dpi. Later treatment was of limited effectiveness for the seasonal IAV strain portrait, but was still effective in the late peaking portrait of infection with an avian IAV strain.

An important distinction between the spatial and non-spatial MMs is that while the latter assumes a spatially uniform virus concentration $V(t)$ that is directly compared against experimental measurements at time $t$, virus concentration in the present MM depends on both time and depth, $V(x, t)$. Herein, $V(t)$ was obtained by averaging $V(x, t)$ over $x \in [0, 30]$ cm to compare against experiments. This is equivalent to assuming that a patient sample, e.g. a nasal wash, takes a somewhat uniform, mixed up sample of the virus concentration in the HRT down to the 30 cm represented in the MM. It is likely that typical experimental measurements sample the concentration over a shallower depth. The impact of this choice on our key findings was explored in S1 Fig. Since advection prevents infection from spreading below the deposition depth, decreasing the depth over which $V(x, t)$ is averaged excludes more of the infection-free regions where $V(x, t) = 0$. This, in turn, slightly increases $V(t)$, the estimate of the virus concentration that would be measured experimentally. In the presence of the full immune response, averaging only over the top 3 cm shortened the time for the virus concentration to fall below 1 $TCID_{50}$/mL by approximately one day. The spatial MM thus offers a unique opportunity to explore how different HRT sampling methodologies might provide a different picture of the infection course and outcome.

The spatial MM presented herein, despite its simplicity, offers a novel and interesting opportunity to study the spatial localization and dissemination of IAV infections within the HRT. In particular, it could serve to test if results obtained by non-spatial ODE MMs are modified when taking into account spatial aspects of IAV infections. In the future, the addition of an explicit, dynamical immune response—to replace the empirical equations used herein as stand-ins for immune response components—would enable the use of this MM to reproduce the kinetics of re-infection, similar to the work done by others [41, 88]. Of particular interest would be to explore the role of IFN secreted by infected cells in inducing an antiviral state in neighbouring cells [50, 51]. This spatial MM would be ideal to explore the effect of diffusion and advection in either enhancing this spatially local role of IFN by spreading it further afield, or decreasing its role by either disseminating its concentration over a larger area or perhaps by allowing IAV to outrun IFN's localized antiviral effect. This has been explored to some extent by others in the presence of diffusion alone [50, 51], but it would be particularly interesting to revisit this process in the presence of both diffusion and upwards advection.

## Methods

### Numerical implementation of the mathematical model

The Euler method was used to numerically solve the cell population equations, namely

$$
\begin{aligned}
T_m^{n+1} &= T_m^n - \Delta t \; \beta T_m^n V_m^n \\
E_{1,m}^{n+1} &= E_{1,m}^n + \Delta t \left[ \beta T_m^n V_m^n - \frac{n_E}{\tau_E} E_{1,m}^n \right] \\
E_{i,m}^{n+1} &= E_{i,m}^n + \Delta t \left[ \frac{n_E}{\tau_E} E_{i-1,m}^n - \frac{n_E}{\tau_E} E_{i,m}^n \right] \qquad i = 2, 3, ..., n_E \\
I_{1,m}^{n+1} &= I_{1,m}^n + \Delta t \left[ \frac{n_E}{\tau_E} E_{n_E,m}^n - \frac{n_I}{\tau_I} I_{1,m}^n \right] \\
I_{j,m}^{n+1} &= I_{j,m}^n + \Delta t \left[ \frac{n_I}{\tau_I} I_{j-1,m}^n - \frac{n_I}{\tau_I} I_{j,m}^n \right] \qquad j = 2, 3, ..., n_I
\end{aligned}
\tag{3}
$$

where $T_m^n = T(x_m, t_n)$, $x_m = m\Delta x$, $t_n = n\Delta t$, and $\Delta x$ and $\Delta t$ are the chosen spatial and temporal

step sizes, respectively. We define $N_x = x_{\max}/(\Delta x) = 3000$, the number of spatial boxes or sites that make up the simulated HRT such that $\Delta x = 0.3$ m/(3000 sites) = 100 μm/site, the diameter of $\sim$4–5 epithelial cells. This $N_x$ was chosen by verifying that choosing a larger number of sites (smaller $\Delta x$) did not affect the solution. When presenting the fraction of target cells or infectious cells, or the virus concentration, as a function of time only, averaged over space, those are computed as

$$T(t) = T^n \quad = \frac{1}{N_x} \sum_{m=1}^{N_x} T_m^n$$

$$I(t) = I^n \quad = \frac{1}{N_x} \sum_{m=1}^{N_x} \sum_{j=1}^{n_I} I_{j,m}^n$$

$$V(t) = V^n \quad = \frac{1}{N_x} \sum_{m=1}^{N_x} V_m^n$$

For the virus concentration, for simplicity, the diffusion and advection terms are each treated separately, and are applied before the production and clearance terms are considered. To solve the diffusion term, the Crank-Nicolson method was used, namely

$$\frac{\partial V(x,t)}{\partial t} = D_{\mathrm{PCF}} \; \frac{\partial^2 V(x,t)}{\partial x^2}$$

$$\frac{V_m^{n+1} - V_m^n}{\Delta t} = \frac{D_{\mathrm{PCF}}}{2} \left[ \frac{(V_{m+1}^{n+1} - 2V_m^{n+1} + V_{m-1}^{n+1}) + (V_{m+1}^n - 2V_m^n + V_{m-1}^n)}{(\Delta x)^2} \right]$$

$$-\alpha V_{m-1}^{n+1} + (2 + 2\alpha) V_m^{n+1} - \alpha V_{m+1}^{n+1} = \alpha V_{m-1}^n + (2 - 2\alpha) V_m^n + \alpha V_{m+1}^n$$

where $\alpha = D_{\mathrm{PCF}} (\Delta x)^2/(\Delta t)$. The rate of viral diffusion in the PCF was estimated based on the Stokes-Einstein equation for IAV diffusing in plasma at body temperature as $D_{\mathrm{PCF}} \approx 10^{-12}$ m$^2$/s [25, 26]. An absorbing boundary condition was used at the top of the HRT, $V(0, t) = 0$ (or $V_0^n = 0$), to capture virus flow out through the mouth and nose. A reflective boundary condition, $V(x_{\max} + \Delta x, t) = V(x_{\max}, t)$ (or $V_{N_x+1}^n = V_{N_x}^n$), was used at the bottom of the HRT. The bottom boundary condition becomes irrelevant once advection is introduced as the flow at $x = x_{\max}$ becomes negligible. With these boundary conditions in place, the diffusion of virus over a step size $\Delta t$ can be expressed as

$$
\begin{bmatrix} V_1^{n+1} \\ V_2^{n+1} \\ \vdots \\ V_{N_x-1}^{n+1} \\ V_{N_x}^{n+1} \end{bmatrix}
=
\begin{bmatrix}
2+2\alpha & -\alpha & 0 & \cdots & 0 \\
-\alpha & 2+2\alpha & -\alpha & \ddots & \vdots \\
0 & \ddots & \ddots & \ddots & 0 \\
\vdots & \ddots & -\alpha & 2+2\alpha & -\alpha \\
0 & \cdots & 0 & -\alpha & 2+\alpha
\end{bmatrix}^{-1}
\begin{bmatrix}
2-2\alpha & \alpha & 0 & \cdots & 0 \\
\alpha & 2-2\alpha & \alpha & \ddots & \vdots \\
0 & \ddots & \ddots & \ddots & 0 \\
\vdots & \ddots & \alpha & 2-2\alpha & \alpha \\
0 & \cdots & 0 & \alpha & 2-\alpha
\end{bmatrix}
\begin{bmatrix} V_1^n \\ V_2^n \\ \vdots \\ V_{N_x-1}^n \\ V_{N_x}^n \end{bmatrix}
$$

The virus advection term can be written as follows

$$\frac{\partial V(x,t)}{\partial t} = v_a \frac{\partial V(x,t)}{\partial x}$$
$$\frac{V_m^{n+1} - V_m^n}{\Delta t} = v_a \frac{V_{m+1}^n - V_m^n}{\Delta x} \tag{4}$$
$$V_m^{n+1} = \left(1 - \frac{v_a \Delta t}{\Delta x}\right) V_m^n + \left(\frac{v_a \Delta t}{\Delta x}\right) V_{m+1}^n .$$

This simplistic numerical scheme is known to lead to a dispersion (diffusion) of the solution [89]. However, the spatial MM simulator imposes $v_a \Delta t/(\Delta x) = 1$, such that Eqn. (4) simplifies to $V_m^{n+1} = V_{m+1}^n$, a trivial translation of the solution, which ensures the latter will remain stable and will not disperse. The speed of the PCF is believed to be approximately the same as that of the mucus blanket and is estimated to be $v_a \approx 40$ μm/s, based on experiments that measured transport in the PCF and mucus layers, using real-time microscopy to observe fluorescent markers in each layer in vertical profile sections of human tracheobronchial epithelial cell cultures [27]. This requires the time step for the spatial MM simulator to be $\Delta t = \Delta x/v_a =$ (100 μm)/(40 μm/s) = 2.5 s. Here, the boundary condition at the top is irrelevant since the solution depends only on the downstream element ($V_1^{n+1} = V_2^n$), and the boundary condition at the bottom is chosen such that $V_{N_x}^n = 0$, i.e. there is no virus beyond the end of the HRT. Once diffusion and advection have been applied, the Euler method is used to solve the remaining terms of the virus equation, namely

$$V_m^{n+1} = V_m^n + \Delta t \left[ p \sum_{j=1}^{n_I} I_{j,m}^n - c V_m^n \right] \tag{5}$$

The droplet-like, initial IAV distribution $V(x, t = 0)$ is represented by a Gaussian centred at the site of deposition,

$$V_m^0 = V(x_m, t = 0) = \frac{V^{0*}}{\sqrt{2\pi\sigma^2}} \exp\left(-\frac{(x_m - x_d)^2}{2\sigma^2}\right) , \tag{6}$$

where $x_d$ is the site of deposition, $\sigma = 0.5$ mm, and $V^{0*}$ is chosen so that $\langle V(x, t = 0) \rangle = \frac{1}{N_x} \sum_{m=1}^{N_x} V_m^0 = 7.5 \times 10^{-2}$ TCID$_{50}$/mL, i.e. the spatial average of the initial virus inoculum concentration in the spatial MM herein is equal to $V(t = 0)$ in the non-spatial ODE MM by Baccam et al. [29].

While most parameters were taken directly from [29], some were adapted. Whereas in Baccam et al. [29] $T + E + I = 4 \times 10^8$ corresponds to the number of cells, in the spatial MM herein, fraction of cells in each state are considered instead such that $T + E + I = 1$. As such, what we consider the Baccam et al. [29] virus production rate is converted from that reported in their paper as

$$p_{\text{Baccam}} = \left(\frac{0.046(\text{TCID}_{50}/\text{mL})}{\text{d} \cdot \text{cell}}\right) \times (4 \times 10^8 \text{cells}) \times \left(\frac{1 \text{ d}}{24 \text{ h}}\right)$$
$$= 7.67 \times 10^5 (\text{TCID}_{50}/\text{mL}) \cdot \text{h}^{-1} .$$

Once advection is introduced (see Results), a value of $11 \times p_{\text{Baccam}}$ is used so that the virus titer will peak at roughly 2–3 dpi, consistent with that observed in Baccam et al. [29].

### Empirical, mathematical representation of the immune response

The time course for interferon (IFN) is represented as

$$F(t) = \frac{2}{e^{-\lambda_g(t-t_p)} + e^{\lambda_d(t-t_p)}} \quad, \tag{7}$$

where $\lambda_g$ and $\lambda_d$ are the growth and decay rates of IFN respectively, and $t_p$ is the time of peak. Since $F(t) \in [0, 1]$, $F(t)$ represents the fractional amount of IFN, relative to peak IFN, at time $t$. In the MM, $\lambda_g = 2$ d$^{-1}$, $\lambda_d = 1$ d$^{-1}$ and $t_p = 3$dpi to match experimental data. Its effect in the MM is to attenuate virus production, $p$, captured as

$$(1 - \varepsilon_{\text{IFN}}) \; p = \left(1 - \frac{F}{F + f_{50}}\right)p, \tag{8}$$

where $f_{50}$ is the amount of IFN required to reduce the virus production rate to one half its normal value, i.e. $p \to \frac{p}{2}$ when $F = f_{50}$. In the MM, $F(t)$ is normalized so as to have a maximum value of 1 ($F_{\max} = 1$). As such, if $f_{50} = 0.1$, then $p \to \frac{p}{2}$ when $F = 0.1 F_{\max} = 0.1$.

The time course of antibodies (Abs) is represented as

$$A(t) = \frac{1}{1 + \left(\frac{1}{A_0} - 1\right)e^{-\alpha t}} \quad, \tag{9}$$

where $A_0$ is the initial amount of Abs, and $\alpha$ is the growth rate of Abs. Since $A(t) \in [0, 1]$, $A(t)$ represents the fractional amount of Abs, relative to peak Abs, at time $t$. In the MM, $\alpha = 0.75$ d$^{-1}$ and $A_0 = 2 \times 10^{-3}$ to match experimental data. The effect of Abs in the MM is to enhance virus clearance, $c$, captured as

$$c + k_A A(t) \tag{10}$$

where $k_A$ (h$^{-1}$) represents the binding affinity of Abs.

The time course of cytotoxic T lymphocytes (CTLs) is represented as

$$C(t) = \frac{2}{e^{-\lambda'_g(t-t'_p)} + e^{\lambda'_d(t-t'_p)}} \quad, \tag{11}$$

where $\lambda'_g$ and $\lambda'_d$ are the growth and decay rates of CTLs, respectively, and $t'_p$ is the time of peak. Since $C(t) \in [0, 1]$, $C(t)$ represents the fractional amount of CTLs, relative to peak CTLs, at time $t$. In the MM, $\lambda'_g = 2$ d$^{-1}$, $\lambda'_d = 0.4$ d$^{-1}$ and $t'_p = 8$ dpi to match experimental data. The effect of CTLs in the MM is to increase the rate of loss of IAV-infected cells which CTLs recognize as infected. Newly infected cells begin to present IAV peptides on their MHC-1 for CTL recognition $\sim 4h$ after IAV infection [17, 54]. In our MM, this corresponds approximately to half the duration of the eclipse phase ($\tau_E/2 = 4h$): cells in $E_{i=[1,n_E/2]}$ would not be presenting peptides, while those in $E_{i=n_E/2+1,n_E}$ would be recognizable and thus could be killed by

CTLs. As such, killing of recognizably-infected cells by CTLs in the MM is captured as

$$
\begin{aligned}
\frac{\partial E_i(x,t)}{\partial t} &= \cdots - 0 & \text{for } i &= 1, 2, ..., \frac{n_E}{2} \\
\frac{\partial E_i(x,t)}{\partial t} &= \cdots - k_C\, C(t)\, E_i(x,t) & \text{for } i &= \frac{n_E}{2} + 1, ..., n_E \\
\frac{\partial I_j(x,t)}{\partial t} &= \cdots - k_C\, C(t)\, I_j(x,t) & \text{for } j &= 1, 2, ..., n_I
\end{aligned}
\tag{12}
$$

where $k_C$ ($\mathrm{h}^{-1}$) provides the scale for the rate of infected cell killing by CTLs.

## Supporting information

**S1 Video. Spatiotemporal course of IAV infection in the presence of diffusion only (no advection).**
(MP4)

**S2 Video. Spatiotemporal course of IAV infection in the presence of diffusion and advection.**
(MP4)

**S3 Video. Same as S2 Video where the fraction of target cells and infectious cells (top row) are shown on a log-scale.** This video better illustrates that, even when looking at cell concentrations down to one in a million ($10^{-6}$), the infection still appears to move downwards along the HRT.
(MP4)

**S1 Fig. Effect of varying the length over which the virus concentration is measured.** The effect of the length, measured from $x = 0$ down to $x = L_{\text{top}}$, over which $V(x, t)$ in the spatial MM is averaged to produce the curves showing Virus versus Time, for variants of the MM (a) without cellular regeneration and a full immune response; (b) with cellular regeneration but without a full immune response; and (c) with cellular regeneration and a full immune response.
(PDF)

**S2 Fig. Effect of the inoculum deposition depth on the complete spatial MM.** Whereas Fig 3(g)–3(i) shows the effect of deposition depth in the earliest version of the MM with diffusion and advection alone, these graphs show the effect of deposition depth of the initial virus inoculum ($x_d$) in the complete spatial MM which includes cellular regeneration and a full immune response.
(PDF)

## Author Contributions

**Conceptualization:** Micaela B. Reddy, Catherine A. A. Beauchemin.

**Formal analysis:** Christian Quirouette, Catherine A. A. Beauchemin.

**Funding acquisition:** Catherine A. A. Beauchemin.

**Investigation:** Christian Quirouette, Nada P. Younis, Catherine A. A. Beauchemin.

**Methodology:** Christian Quirouette, Catherine A. A. Beauchemin.

**Project administration:** Catherine A. A. Beauchemin.

**Software:** Christian Quirouette, Catherine A. A. Beauchemin.

**Supervision:** Catherine A. A. Beauchemin.

**Visualization:** Christian Quirouette, Catherine A. A. Beauchemin.

**Writing – original draft:** Christian Quirouette, Catherine A. A. Beauchemin.

**Writing – review & editing:** Christian Quirouette, Micaela B. Reddy, Catherine A. A. Beauchemin.

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
