## [Decision Letter · Decision Letter 0]

20 Oct 2019

Dear Dr Quirouette,

Thank you very much for submitting your manuscript, 'A mathematical model describing the localization and spread of influenza A virus infection within the human respiratory tract', to PLOS Computational Biology. As with all papers submitted to the journal, yours was fully evaluated by the PLOS Computational Biology editorial team, and in this case, by independent peer reviewers. The reviewers appreciated the attention to an important topic but identified some aspects of the manuscript that should be improved.

We would therefore like to ask you to modify the manuscript according to the review recommendations before we can consider your manuscript for acceptance. Your revisions should address the specific points made by each reviewer and we encourage you to respond to particular issues Please note while forming your response, if your article is accepted, you may have the opportunity to make the peer review history publicly available. The record will include editor decision letters (with reviews) and your responses to reviewer comments. If eligible, we will contact you to opt in or out.raised.

- Supporting Information uploaded as separate files, titled 'Dataset', 'Figure', 'Table', 'Text', 'Protocol', 'Audio', or 'Video'.

We hope to receive your revised manuscript within the next 30 days. If you anticipate any delay in its return, we ask that you let us know the expected resubmission date by email at ploscompbiol@plos.org.

Sincerely,

Rustom Antia

Associate Editor

PLOS Computational Biology

Rob De Boer

Deputy Editor

PLOS Computational Biology

[LINK]

Comments from Associate Editor (RA):

I enjoyed reading the paper.  The work certainly tackles an important problem -- the effect of spatial considerations on the dynamics of infection. I like the way the paper has addressed an important aspect of this problem.  There is one aspect that I believe could be expanded on -- the  assumptions of the model.  Reviewer 1 has I think gone into this in detail and I think addressing these would substantially facilitate others in building on your very nice paper. 

Reviewer's Responses to Questions

**Comments to the Authors:**

Reviewer #1: Quirouette et al. studies influenza infection within a host using a mathematical modeling approach. Different from most previous modeling works, the authors took a partial-differential equation approach and explicitly considered the spatial aspect of the infection. More specifically, the authors used the model to evaluate the roles of diffusions and advection of free viral particles on spatial spread of infection, and conclude that the advection plays an important role in moving infection towards upper human respiratory tract (HRT). The authors then analyzed variations of the model to evaluate the impacts of target cell regeneration, immune responses (including the interferon (IFN), the antibody and the cytotoxic T lymphocyte (CTL) response) on infection dynamics. Finally, the authors used the model to explain the differences between seasonal influenza and avian influenza infections and the effectiveness of antivirals against them.

Influenza infection process is inherently spatial, especially during the initial period of infection. Despite this fact, very few theoretical works have been done to understand how spatial structure impacts on infection dynamics and their interaction with the immune responses [Gallagher, et al. Viruses 2018; 10(11),627]. This work directly addresses this important issue and provides novel insights into the role of spatial structure during influenza infection. In addition, the model developed in this work serves a good baseline model that can be potentially adapted by researchers to understand the spatial infection process for other respiratory. The analysis of this study is rigorous, and the manuscript is well written. I think it is suitable for publication; however, I have some major and minor concerns with the assumptions of the model that are listed below and shall be addressed appropriately.

First of all, one assumption I found puzzling is that the infection starts at 15cm (in the middle) of the HRT. Is it consistent with the initial infection site of a typical influenza infection? The authors showed that when the position of this initial infection site is higher (x is smaller than 15cm), the model predicts only a very small fraction of target cells are infected due to the high advection rate (Fig. 3 j-l and Fig. 4). I think this is a prediction inconsistent with an actual influenza infection, where the initial infection site is higher in the HRT than assumed in the model, and the establishment of infection shall not be very sensitive to where it starts. I think it is important to use a more biologically plausible value for where the infection starts at the HRT in simulations.

My interpretation of the inconsistency above is that other parameters used in the model may be unrealistic. For example, the advection rate in the work (40 mm/s) is taken from estimates of the advection rate for periciliary liquid (ref. 47 within the manuscript). However, it is highly likely that the viral particles get tracked in the tissue while they travel with the periciliary liquid, leading to a smaller advection rate for viruses. It is not unreasonable to speculate that the advection rate for viruses would be an order of magnitude smaller than the periciliary liquid due to the complicated structure of the epithelium cells on the surface of HRT. Therefore, given the uncertainty in the advection rate, it is crucial to evaluate the robustness of all the results against variations in the advection rate.

Second, the authors modeled the impact of the interferon response as a reduction in the viral production rate (p). However, interferon response also protects susceptible cells from infection through paracrine signaling. This can have a strong impact on the spread of viruses especially the spread is spatial (Huang, et al. Frontiers in Immunology 2019; 10, 1736; Domingo-Calap et al. Nat Microbiol. (2019) 4:1006–13.). Therefore, the conclusions about the role of the interferon response during spatial spread in this work are likely to be incorrect/incomplete. I understand extending the model explicitly with an interferon component may go beyond the scope of the study. In that case, I think the authors shall either avoid modeling the interferon response or thoroughly discuss the limitations of current formulation in the context of the existing literature.

Third, maybe this is not intended by the authors; but the current manuscript reads like that the spatial spread of the influenza virus as modeled using the PDEs is the only mode of spread. In reality, it is likely that the virus spread both locally (with a spatial structure) and over long distances (without a spatial structure), especially when the infection becomes systematic. I think this is one of the reasons that the previous ODE models are useful in addressing many questions, despite the spatial structure in the spread. I think the work is interesting in that it addresses the spatial issue and sheds light on the spatial aspect of the infection dynamics; at the same time, it is important to fully discuss the limitations of the PDE model and acknowledge the infection process modeled is one aspect of infection. It’d be useful to have a paragraph discussing when the PDE model is or is not appropriate/needed to model the influenza infection dynamics.

Fourth, I found that the section on comparing seasonal and avian influenza infections and the subsequent predictions of the impact of antivirals on these infections unconvincing. The authors adjusted model parameters such that the infection dynamics are ‘somewhat’ consistent with experimental data (Fig. 10). However, the authors provide no evidence demonstrating that the differences in seasonal and avian influenza are indeed because of the differences in the parameter values of the model. Thus, the predictions of the impact of antivirals on infection dynamics should be treated as hypotheses rather than conclusions.

Fifth, (a minor comment) I find the length of the manuscript is too long. It’d be better if the authors try to reduce the length of the manuscript and the number of figures, and highlights the most important aspects of the results.

Reviewer #2: This paper by Quirouette et al describes a highly novel PDE spatiotemporal of model of influenza infection. The paper is extremely clearly written. Stepwise additions to the model (advection, target cell replenishment, immunity) are performed in a logical manner. The figures and movies generally illustrate the key concepts well. The model breaks new conceptual ground in describing the spatial pathogenesis of influenza respiratory tract infection and by demonstrating that the standard viral dynamics model misses this key component of infection dynamics. I am also persuaded by the argument that the simple PDE approach is preferable to an agent-based model for capturing these effects given the nature of available data. The depth of literature review informing the model & validating its conclusions is also impressive. Overall, this is a thoughtfully done, important paper.

I have a number of minor comments to enhance the clarity of scientific messaging:

1. While the abstract makes sense after reviewing the entire paper, it confused me on first read. The mucus escalator and advection (not commonly used terms for modeling or virology generalists) are mentioned in the first sentence prior to being formally defined. The phrase “IAV advection dominates kinetics” is vague. It would be better to say that advection rather than diffusion is necessary to explain existing quantitative data as well as empirically derived qualitative spatial features of IAV infection. The sentence regarding antiviral therapy does not point to any particular scientific conclusion. The existence of experimental data consistent with model predictions is left unmentioned.

2. While the paper generally does a great job of reviewing existing IAV modeling literature and concepts incorporated into the model (the sections on immunity and cellular regeneration are particularly useful for the reader), the introductory paragraph regarding the mucous escalator (lines 14-16) is rather sparse. Only one paper is cited (a NEJM review which focuses mostly on the biophysical and biochemical properties of mucous). I would appreciate more detail regarding the role of cilia, the overall purpose of mucous clearance, and an explanation for why viral particles are likely to have a similar advection rate to the surrounding mucous. Specifically, the data used to inform parameter values from reference 47 (Table 1, line 57) should be described in some detail as the advection rate is central to all model conclusions. Similar detailed attention should be given to describing the diffusion rate derived from ref 37.

3. Line 60: the assumption that virus bounces back or is lost at the lower end of the human respiratory tract seems unlikely. Pneumonitis / lower tract disease is one of the feared complications of IAV and in conjunction with bacterial superinfection accounts for most IAV related mortality.

4. Figure 2 & figure 3 d-f: it is impossible to tell the lines apart at higher dpi.

5. Line 102: how does the new viral production rate (11 * pbaccam) compare to in vitro estimates?

6. Lines 115-16: Fig 3g-i makes an interesting prediction. However, I disagree with the strength of the conclusion that depth >15 cm has little effect as even a 0.5 log increase in peak viral load may be of clinical significance.

7. Lines 120-122: I wonder if this conclusion would hold with a stochastic formulation of the model. An interesting feature of a stochastic version of the model may be to predict whether deeper viral site of initial infection is more predictive of successful transmission / acquisition.

8. Lines 134-135: this prediction that infection moves from lower to upper tract despite viral load doing the opposite is a key interesting prediction of the model. However, S2 video only captures this concept somewhat. I suggest creating another video in which the y-axis for fraction of target & infected cells is log-10 converted so that the difference in # of infected cells between the top and bottom of the respiratory tract at early timepoints is more evident.

9. Figure 4 is very instructive. However, I am left wondering why diffusion is left in the model at all if it is not needed to recapitulate the key kinetic features of observed data.

10. Figure 9: these in silico knock out experiments are novel and justify what I would call a rather complex model of immunity relative to most in the field (not simplistic as stated by the authors). However, the authors should mention in the discussion that there are large limitations in terms of extrapolating conclusions from murine models of viral infection to mathematical models intended to capture kinetics of human infection. Lines 264-268 state this well and should be restated as a limitation in the discussion.

11. Lines 384-386: I may be missing the point, but it seems to me that this underestimate is related to per cell viral production. The model is trained to total measured virus so it is less clear how total viral load is underestimated. Is the idea that the lower tract is not sampled and therefore current clinical studies underestimate total viral load?

12. Lines 392-6: as mentioned before, fatal lower tract (lung) disease occurs in a minority of infected persons and is a particular issue in those with T cell immunodeficiency. This argues for some role for diffusion in viral seeding of more distal airways.

13. Lines 406-408, 422-426: These 2 studies showing the trafficking pattern of fluorescent IAV and quantifying % of cells killed during IAV infection are really important for complementing and validating the model’s predictions. Until I reached these paragraphs, my impression was that the paper was purely theoretical and hypothesis generating. However, these experiments provide greater weight that the model’s central qualitative conclusions are likely to be true. As such, I think it would be fair and useful to mention these papers earlier in the paper, both in the introduction and abstract. I also think it would be great to include a paragraph describing experiments that the authors think would further validate and enhance the model.

14. Given the uncertain early kinetics of H5N1 described by the authors, the modeling of H5N1 seems a bit speculative. Moreover, this section of the paper does not add to the overall premise of the paper, which is that advection is fundamental for understanding the temporo-spatial kinetics of IAV. It seems unlikely that advection would only be important for some but not other strains of IAV.

**Have all data underlying the figures and results presented in the manuscript been provided?**

Reviewer #1: Yes

Reviewer #2: Yes

PLOS authors have the option to publish the peer review history of their article (what does this mean?). If published, this will include your full peer review and any attached files.

Reviewer #1: No

Reviewer #2: Yes: Joshua T Schiffer

---

## [Decision Letter · Decision Letter 1]

31 Jan 2020

Dear Mr. Quirouette,

We are pleased to inform you that your manuscript 'A mathematical model describing the localization and spread of influenza A virus infection within the human respiratory tract' has been provisionally accepted for publication in PLOS Computational Biology.

Before your manuscript can be formally accepted you will need to complete some formatting changes, which you will receive in a follow up email. A member of our team will be in touch within two working days with a set of requests.

Best regards,

Rustom Antia

Associate Editor

PLOS Computational Biology

Rob De Boer

Deputy Editor

PLOS Computational Biology

Reviewer's Responses to Questions

**Comments to the Authors:**

Reviewer #1: I think the revisions are acceptable.

**Have all data underlying the figures and results presented in the manuscript been provided?**

Reviewer #1: None

PLOS authors have the option to publish the peer review history of their article (what does this mean?). If published, this will include your full peer review and any attached files.

Reviewer #1: Yes: Ruian Ke

---

## [Editor Report · Acceptance letter]

25 Mar 2020

PCOMPBIOL-D-19-01437R1 

A mathematical model describing the localization and spread of influenza A virus infection within the human respiratory tract

Dear Dr Quirouette,

I am pleased to inform you that your manuscript has been formally accepted for publication in PLOS Computational Biology. Your manuscript is now with our production department and you will be notified of the publication date in due course.

With kind regards,

Laura Mallard
